# Landscape of submitochondrial protein distribution

F.-Nora Vögtle[1], Julia M. Burkhart[2], Humberto Gonczarowska-Jorge[2], Cansu Kücükköse[1,3], Asli Aras Taskin[1], Dominik Kopczynski[2], Robert Ahrends[2], Dirk Mossmann[1], Albert Sickmann[2,4,5], René P. Zahedi[2] & Chris Meisinger[1,6]

The mitochondrial proteome comprises ~1000 (yeast)–1500 (human) different proteins, which are distributed into four different subcompartments. The sublocalization of these proteins within the organelle in most cases remains poorly defined. Here we describe an integrated approach combining stable isotope labeling, various protein enrichment and extraction strategies and quantitative mass spectrometry to produce a quantitative map of submitochondrial protein distribution in *S. cerevisiae*. This quantitative landscape enables a proteome-wide classification of 986 proteins into soluble, peripheral, and integral mitochondrial membrane proteins, and the assignment of 818 proteins into the four subcompartments: outer membrane, inner membrane, intermembrane space, or matrix. We also identified 206 proteins that were not previously annotated as localized to mitochondria. Furthermore, the protease Prd1, misannotated as intermembrane space protein, could be re-assigned and characterized as a presequence peptide degrading enzyme in the matrix.

[1] Institute of Biochemistry and Molecular Biology, ZBMZ, Faculty of Medicine, University of Freiburg, Freiburg im Breisgau 79104, Germany. [2] Leibniz-Institut für Analytische Wissenschaften-ISAS-e.V, Dortmund 44139, Germany. [3] Faculty of Biology, University of Freiburg, Freiburg im Breisgau 79104, Germany. [4] Department of Chemistry, College of Physical Sciences, University of Aberdeen, Aberdeen AB24 3FX, UK. [5] Medizinisches Proteom Center, Ruhr Universität Bochum, Bochum 44801, Germany. [6] BIOSS Centre for Biological Signalling Studies, University of Freiburg, Freiburg im Breisgau 79104, Germany. F.-Nora Vögtle and Julia M. Burkhart contributed equally to this work. Correspondence and requests for materials should be addressed to R.P.Z. (email: zahedi@isas.de) or to C.M. (email: chris.meisinger@biochemie.uni-freiburg.de)

In the past decade extensive proteomic studies of purified organellar fractions have generated large high quality compendia of mitochondrial proteins for various organisms ranging from 850 proteins in yeast to more than 1500 proteins in human[1–7]. However, knowledge about specific protein sublocalization to one of the four subcompartments, the outer (OM) and inner membrane (IM) and the two soluble compartments intermembrane space (IMS) and matrix, in which these proteins fulfill their dedicated tasks, is limited. Indeed, even for well-established mitochondrial proteins data on submitochondrial localization currently available in the standard databases[8, 9] are often incorrect or incomplete. This might partially result from the incorporation of bioinformatic predictions without clear experimental support or the direct inclusion of high-throughput data that often bears misannotations. Large scale tagging approaches for subcellular and suborganellar localization are particularly problematic in case of mitochondria, because tags often interfere with the protein import and sorting machineries, either preventing import or leading to mislocalization of tagged proteins[6, 10]. E.g., (i) one of the most classical mitochondrial matrix proteins, citrate synthase (Cit1) is not annotated as matrix protein in the yeast genome database (SGD) while Uniprot localizes it to the matrix but also the IMS and the cytosol[8, 9]. (ii) In 2002, De Hertogh et al.[11] predicted a large number of integral yeast membrane proteins based on phylogenetic classifications and to date all of these proteins are annotated, e.g., in the yeast genome database, as integral membrane proteins[8]. This includes many mitochondrial proteins, such as Tim9, Tim10, and Tim13, which are prime examples of soluble proteins of the IMS[5, 12], or the peripheral IM subunits of the respiratory complex IV (Cox4 and Cox12) and the ATP-synthase (Atp1, Atp2, or Atp3), all clearly not integral membrane proteins[13]. Moreover, the APEX approach recently introduced by the Ting lab, which elegantly deciphers proteins in the proximity of a suborganellar targeted bait protein, allows deduction of these proteins' toponomes but falls short on deciphering their clear assignment to one of the four subcompartments[14, 15]. Here, we generate a high quality map of submitochondrial distribution for native proteins under clearly defined experimental conditions.

## Results

**Quantitative mapping of mitochondrial proteins.** The first part of our experimental approach is based on the dissection and global identification of soluble, peripheral, and integral membrane proteins from mitochondria of the model system yeast. We used stable isotope labeled yeast cultures (SILAC) and isolated highly pure light and heavy mitochondria[16–18] (Fig. 1a; Supplementary Fig. 1). Separation of the three protein classes was achieved by either (i) carbonate treatment at pH 11.5 to separate integral membrane proteins (retained in the pellet, $PEL_{carb}$) from peripheral membrane and soluble proteins (extracted into the supernatant, $SN_{carb}$)[19] or (ii) sonication of the purified organelles to separate a complete membrane fraction (incl. integral and peripheral proteins; $PEL_{son}$) from the soluble proteins ($SN_{son}$) (Fig. 1b). Resultant pellet and supernatant samples from light and respective heavy fractions were mixed and SILAC ratios were determined by quantitative mass spectrometry from four replicates (two biological replicates with each time one forward (e.g., $SN_{heavy}/PEL_{light}$) and one reverse experiment (e.g., $SN_{light}/PEL_{heavy}$) respectively; Fig. 1a) quantifying 1053 different proteins. The determined $SN_{carb}/PEL_{carb}$ and $SN_{son}/PEL_{son}$ ratios (Supplementary Data 2) allowed the classification of 805 proteins into integral membrane, peripheral membrane and soluble proteins (Fig. 1a, b). We plotted these ratios ($SN_{carb}/PEL_{carb}$ ($y$-axis) and $SN_{son}/PEL_{son}$ ($x$-axis)), searched for well-known and experimentally verified reference proteins (reference set) from each class and found that they

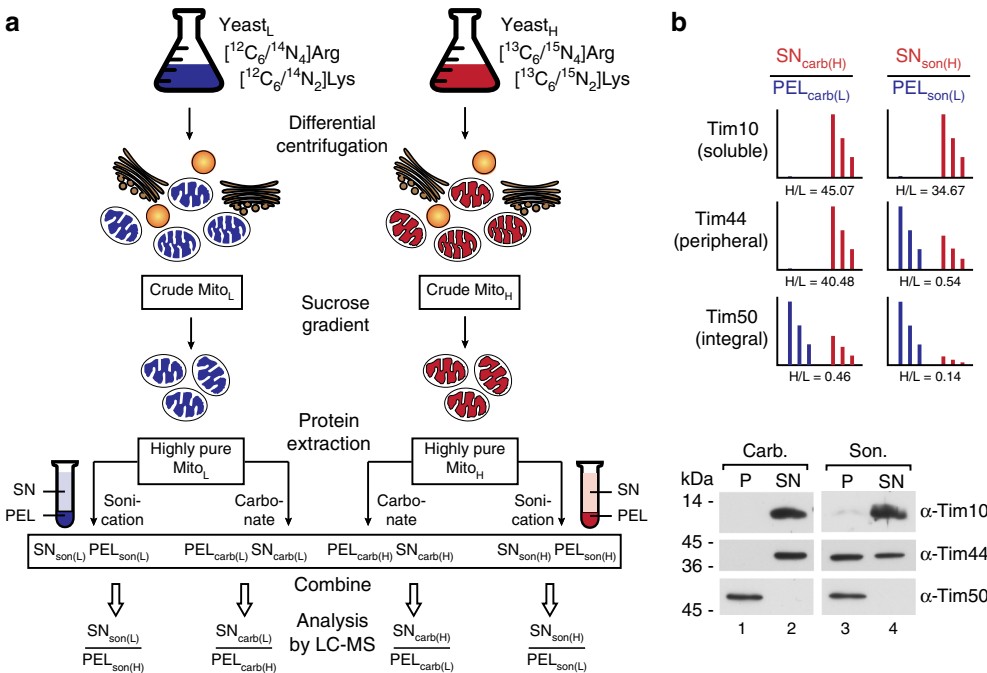

**Fig. 1** Experimental design for global profiling of peripheral membrane, integral membrane, and soluble proteins in highly pure yeast mitochondria. **a** Highly pure mitochondria were isolated from yeast cultures grown in the presence of either "light" (L) or "heavy" (H) amino acids (Arg, Lys). Protein extraction was performed based on sonication to separate membrane proteins ($PEL_{son}$) from soluble proteins ($SN_{son}$) or carbonate treatment, which results in a pellet containing integral membrane proteins ($PEL_{carb}$) and a supernatant with peripheral membrane and soluble proteins ($SN_{carb}$). Respective SN/PEL SILAC ratios were determined by LC-MS. *carb*, carbonate; *son*, sonication. **b** Exemplary SILAC results **a** for Tim proteins representing the three different protein classes analyzed, confirmed by immunoblotting

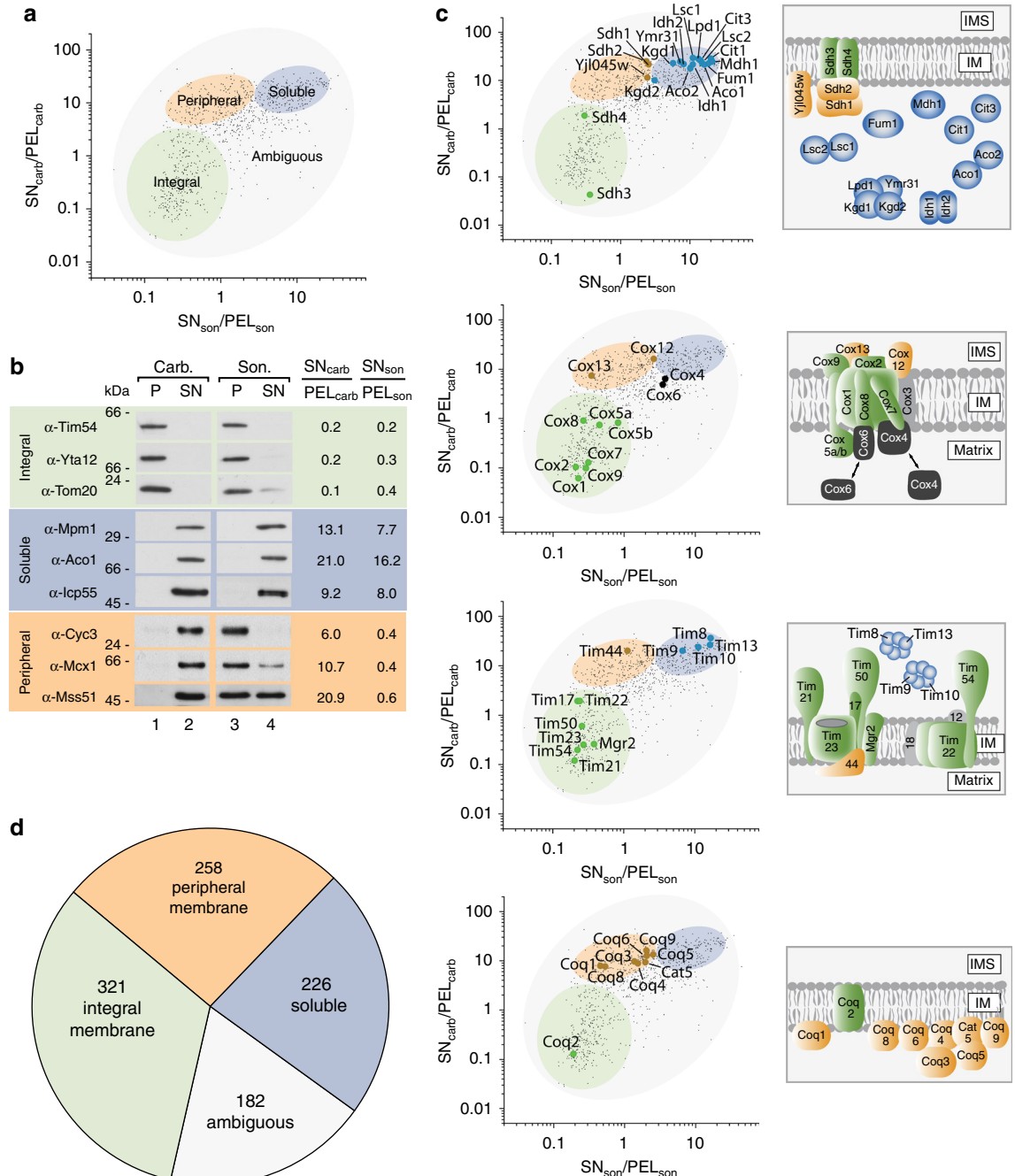

**Fig. 2** Mapping of mitochondrial proteins into different classes. **a** Separation of the mitochondrial proteome into an integral membrane, a peripheral membrane, a soluble, and an ambiguous fraction. **b** Validation of map positions for several mitochondrial proteins by immunoblotting. **c** Map data correlation with known submitochondrial localization of components of the translocase of the inner membrane (*TIM*), the citrate cycle machinery, complex IV of the respiratory chain, and the Coenzyme Q biosynthesis apparatus[5, 13, 24–26]. Proteins in *gray* could not be mapped by LC-MS. **d** Distribution of the mitochondrial proteome into the indicated classes

cluster in clearly distinguishable clouds (Supplementary Fig. 2, Fig. 2a, searchable Supplementary Data 1) which could be statistically verified based on a mixed multivariate normal distribution model (Supplementary Figs. 3 and 4; see also Methods section). The correlation between the assigned clusters and the actual biochemical separation was verified by immunoblotting, as depicted in Fig. 2b. Notably, 182 proteins appear in an intermediate (ambiguous) region (Fig. 2a; Supplementary Data 1). These proteins represent, e.g., dually localized proteins, such as Mcr1, which exists in an integral OM and a soluble IMS form[20] or several ribosomal subunits, which may switch between a

soluble and an IM associated form (Supplementary Fig. 5a, b)[21]. The SN/PEL values of these proteins represent hybrid values from two differently localized protein pools and are therefore clustering in the ambiguous region. Also the entire set of cytosolic glycolysis enzymes is found in this region reflecting their unique interaction with the outer mitochondrial membrane[22]. Similarly, Mdv1, a component of the mitochondrial fission machinery that associates from the cytosolic site with the OM[23] appeared in this ambiguous region (Supplementary Fig. 5b).

To further validate how precisely this map reflects the actual protein classes we chose several well-known mitochondrial

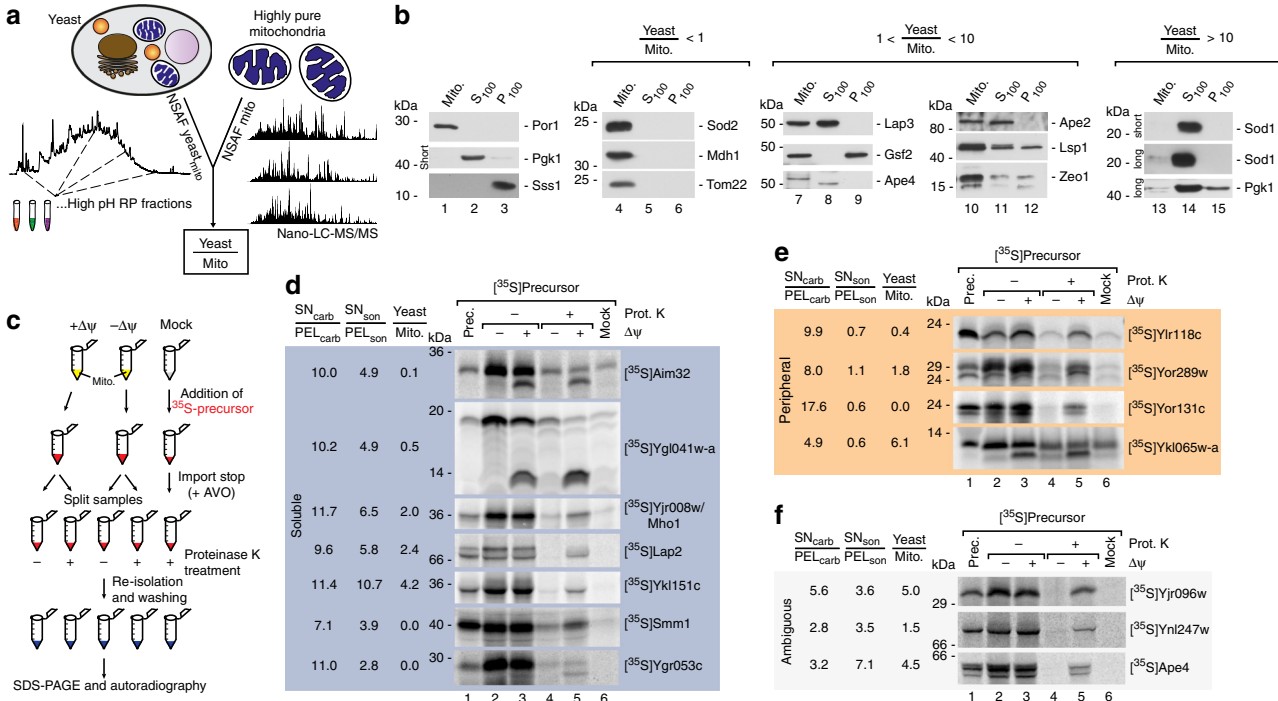

**Fig. 3** Validation of mitochondrial localization for novel proteins. **a** Quantitative assessment of protein abundance in total yeast cells and purified mitochondria based on spectral counting (normalized abundance factor). Resulting Yeast/Mito ratios indicate presence of proteins either exclusively in mitochondria or in multiple cellular compartments. **b** Immunoblot analysis of cellular fractions containing pure mitochondria (Mito), enriched cytosolic proteins (S100), or microsomal proteins (P100). Por1, mitochondrial marker; Pgk1, cytosolic marker; Sss1, ER marker. long, long exposure time; short, short exposure time. **c** Schematic overview of in organello import reactions to validate mitochondrial localization. [$^{35}$S]labeled precursors of candidate proteins were generated by in vitro transcription/translation and incubated with isolated mitochondria in the presence or absence of the membrane potential ($\Delta\psi$) or without mitochondria (Mock). Import reaction was terminated by depletion of the membrane potential and samples were treated with Proteinase K where indicated to remove non-imported precursors. Samples were analyzed by SDS–PAGE and radiolabelled proteins visualized by autoradiography. In case of presequence cleavage upon import a size shift from the precursor to the mature protein can be observed. **d** In organello import of indicated radiolabelled precursors of novel mitochondrial proteins which localize to the soluble protein fraction. **e** In organello import of indicated radiolabelled precursors of novel mitochondrial proteins which localize to the peripheral membrane protein fraction. **f** In organello import of protein candidates from the ambiguous fraction. Prot. K, Proteinase K; prec. precursor; $\Delta\psi$, membrane potential across the inner mitochondrial membrane

protein machineries and complexes with established topologies comprising proteins from all three classes, i.e., soluble, peripheral, and integral membrane proteins (Fig. 2 and Supplementary Fig. 6). These included the preprotein translocase of the IM (TIM), the entire set of citrate cycle enzymes, Complex IV of the respiratory chain and the Coenzyme Q (CoQ) biosynthesis apparatus[5, 13, 24–26]. Indeed, all identified proteins of these complexes localized to the expected clusters in our map (Fig. 2c). For the CoQ machinery a clear separation of the integral Coq2 protein from the remaining peripheral membrane proteins could be observed. Interestingly, the remaining CoQ proteins cluster in two distinct sets in the peripheral membrane protein cloud which might indicate a pool of more tightly associated membrane proteins (Coq1 and Coq8) and a more loosely associated pool (Fig. 2c). Notably, two components of Complex IV, Cox4 and Cox6, were found in the ambiguous region – indeed, it has been shown that both co-exist in a peripheral membrane bound as well as a soluble form[27]. This dual localization consequently leads to their appearance in the ambiguous region of our map (Fig. 2c).

In summary, we identified 321 integral membrane proteins, 258 peripheral membrane proteins, 226 soluble proteins, and 182 ambiguous proteins (Fig. 2d).

**Identification of novel mitochondrial proteins.** From the proteins identified in our data set 206 were not yet annotated as mitochondrial proteins in neither the SGD database nor the PROMITO data set (representing the so far largest compendium of yeast mitochondrial proteins[6–8]) and might therefore represent novel mitochondrial proteins (Supplementary Data 3). Owing to the high sensitivity of mass spectrometry some of these identifications might result from contaminations from other cellular compartments. Others might have escaped previous detection due to low mitochondrial abundance or localization in multiple cellular compartments, including mitochondria. We therefore compared their relative abundance within mitochondria and complete yeast cells, based on label free spectral counting and determined Yeast/Mito ratios for all mitochondrial proteins (Fig. 3a). The majority of known mitochondrial proteins, but also many of the novel candidates, showed low Yeast/Mito ratios indicating a strong enrichment in the mitochondrial fraction (Supplementary Data 2). In contrast, several proteins revealed high Yeast/Mito ratios pointing towards multiple subcellular localizations or a localization in another cellular compartment besides mitochondria. E.g., the dually localized cytosolic and mitochondrial protein superoxide dismutase 1 (Sod1) showed a Yeast/Mito ratio of 16.6 while the exclusively mitochondrial localized superoxide dismutase 2 (Sod2) had a ratio of 0.9 (Supplementary Data 2). Similarly, the OM associated Mito-ER-cortex anchor Num1 had a ratio of 14.5 and the cytoplasmic and mitochondrial localized alanyl-tRNA synthetase Ala1 appeared with a ratio of 5.6. We generated antibodies against several proteins that appeared with higher Yeast/Mito ratios including the novel mitochondrial candidate protein Ape4

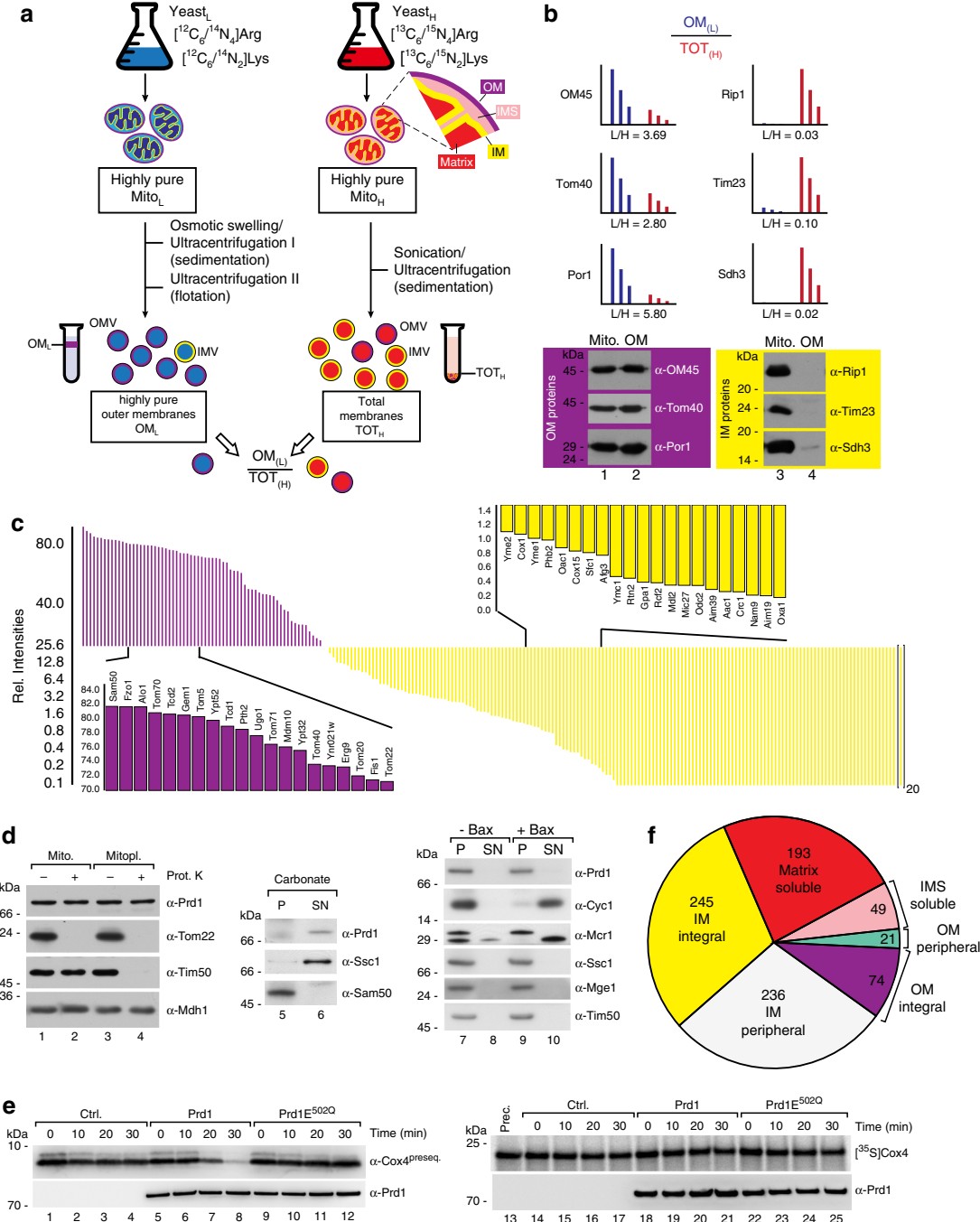

**Fig. 4** Integrating the landscape of submitochondrial protein distribution. **a** Experimental design for global allocation of integral outer and inner membrane proteins. Highly purified mitochondria from a light yeast culture were used for isolation of highly pure outer membrane fractions[31] (OM_L). Total membrane fractions were generated from mitochondria of a heavy labeled yeast culture (TOT_H). Both samples were mixed and the ratios OM_L/TOT_H measured by LC-MS. **b** OM_L/TOT_H ratios of selected integral outer and inner membrane proteins and validation by immunoblotting. **c** Array of determined OM_L/TOT_H ratios reveals distribution of integral mitochondrial outer and inner membrane proteins (shown as relative intensities for proteins in the OM fraction). **d** Sublocalization of Prd1 to the mitochondrial matrix. **e** Cell-free translated Prd1 and Prd1^E502Q protein was incubated with Cox4 presequence peptide (Cox4^preseq.; *left* panels) or radiolabeled Cox4 precursor protein (*right* panels) for indicated time. Ctrl., control (mock translation). **f** Complete overview of the landscape of submitochondrial protein distribution. Allocation of the mitochondrial proteome to the various subcompartements deciphered in this study

(Supplementary Data 3), that has been annotated as a cytoplasmic aspartyl aminopeptidase[8, 28] and tested their localization in different cellular compartments by immunoblotting. While proteins with Yeast/Mito ratios <1 were detected only in the mitochondrial fraction (shown for the resident mitochondrial proteins, Sod2, Mdh1 and Tom22; Fig. 3b, lanes 4–5) proteins with ratios that range between 1 and 10 are present in additional cellular subfractions containing cytosol (S100) or microsomes (P100; Fig. 3b, lanes 7–12). Proteins with ratios >10 can be traced to mitochondria only after long exposure of the immunoblot as shown, e.g., for the eclipsed distributed superoxid dismutase Sod1 (Fig. 3b, lanes 13–14).

To validate the mitochondrial localization of further novel identified proteins we employed in organello import assays. For this we generated radiolabelled precursor proteins by in vitro transcription/translation and successfully labeled proteins were imported into isolated mitochondria[6, 29]. Typically, preprotein import into mitochondria—particularly of IM and matrix proteins —depends on the membrane potential ($\Delta\psi$) across the IM. For many of the mitochondrial precursors an N-terminal presequence serves as targeting signal and is removed by the matrix processing protease (MPP) upon import[29]. Both, $\Delta\psi$-dependency and presequence processing are bona fide criteria for mitochondrial localization[6, 29, 30] and can be monitored after in organello import reactions via sodium dodecyl sulfate polyacrylamide gel electrophoresis (SDS–PAGE) and autoradiography (Fig. 3c). For in organello import we tested 14 different preproteins, which were so far not annotated as mitochondrial proteins and appeared as either soluble or peripheral membrane or ambiguous proteins in our map. All preproteins are imported into mitochondria dependent on the presence of the membrane potential $\Delta\psi$ (Fig. 3d–f) unveiling them as mitochondrial proteins. For several preproteins an additional size shift is observed indicating presequence cleavage by MPP at the matrix site. For some of the tested preproteins we also observed high Yeast/Mito ratios (e.g., Ykl151c with 4.2 or Ykl065w-a with 6.1), which points to a localization to additional cell compartments besides mitochondria. Taken together, we identified 206 proteins that were not assigned to mitochondria so far and biochemically confirmed mitochondrial localization for 14 of these.

**Deciphering integral inner and outer membrane proteins**. To further refine our global map of submitochondrial protein distribution we experimentally assigned the above identified integral membrane proteins to the outer and IM, respectively. While outer mitochondrial membrane vesicles can be purified to a high degree, the biochemical isolation of pure IMs has proven difficult. Therefore, thus far global profiling of the protein composition was performed for the outer but not the IM[31–34] and ~80 resident OM proteins have been identified in yeast[31]. To globally decipher integral IM proteins, we used SILAC to determine the amount of integral membrane proteins present in carbonate resistant pellet fractions from highly pure OMs ($OM_L$) compared to total mitochondrial membranes ($TOT_H$) (Supplementary Data 4 and Fig. 4a). From the obtained $OM_L/TOT_H$ ratio enrichment of proteins in the OM fraction can be distinguished from depletion, the latter being indicative of an IM protein (Fig. 4b, c and Supplementary Fig. 7). In total, this analysis identified 171 different integral IM proteins and 66 integral OM proteins (Fig. 4c and Supplementary Data 4) which can be integrated now into the mitochondrial protein distribution map.

**Integrating a map of submitochondrial protein distribution**. While the proteomes of the two smaller mitochondrial subcompartments, the OM and IMS, had been deciphered with high accuracy and coverages of more than 85% (80 resident proteins[31]) and 90% (50 proteins[17]) in yeast, the protein compendia of the IM and the matrix, expected to be many times larger, could not be resolved yet. We used the well-established OM and IMS reference proteomes[17, 31] (Methods section) to deduce the sublocalization of all mitochondrial proteins characterized above and to generate a landscape of submitochondrial protein distribution (Fig. 4f; Supplementary Data 5). This includes also the assignment of 237 integral membrane proteins to either the outer or the IM (Supplementary Data 4). In total our landscape of submitochondrial protein distribution comprises 236 peripheral and 245 integral IM

proteins, 21 peripheral and 74 integral OM proteins, 49 IMS and 193 soluble matrix proteins (Fig. 4f; Supplementary Data 5 (165 proteins were assigned as ambiguous)).

Manual inspection of our landscape revealed several conflicts with previously annotated sublocalization data. E.g., Pet191 was reported as an integral IM protein[8, 9, 35] while our landscape clearly assigns it as soluble, consistent with our previous localization of the protein in the IMS proteome[17]. A further example is the protease Cym1, which is annotated as IMS protein in the databases[8, 9]. In our landscape it is assigned as soluble matrix protein and indeed was also biochemically verified as matrix localized[36]. The proteins Tim9, Tim10, and Tim13, which were annotated as integral membrane proteins in the yeast genome database (based on phylogenetic predictions; see above)[8, 11] were found as soluble proteins of the IMS (Figs. 1b and 2c, Supplementary Data 1 and 5). A further example is Mpm1, which has been reported as carbonate resistant IM protein[37], but appeared as a highly soluble protein in our map. By using a specific antiserum against Mpm1, we could verify that upon both carbonate treatment and sonication it is completely present in the supernatants (Fig. 2b; Supplementary Data 1 and 5).

Another example of a striking conflict was the reported IMS localization of the metalloprotease Prd1[8, 9, 38]. While the protease was not detected in the IMS proteome[17], it behaved clearly like a soluble protein in this study (Supplementary Data 1 and 5). According to our criteria described above, we therefore expected Prd1 as mitochondrial matrix protein. We generated specific antisera and tested if Prd1 behaves like a matrix protein upon access of Proteinase K to the different subcompartments using mitochondria, which were treated with iso- or hypoosmotic buffers (Fig. 4d). While isoosmotic conditions allow Proteinase K to degrade only OM proteins accessible from the cytosolic site, hypoosmotic conditions enable the protease to degrade proteins in or exposed to the IMS[17]. Prd1 was unaffected under both conditions, clearly confirming it as a soluble matrix protein (Fig. 4d, lane 1–6). Moreover, we used an independent approach to show that Prd1 is not localized to the IMS: incubation of isolated yeast mitochondria with the protein Bax leads to the release of the entire set of soluble proteins from the IMS[17]. However, upon treatment with Bax, Prd1 remained entirely within mitochondria unlike cytochrome *c* or the IMS form of Mcr1, which are efficiently released to the supernatant (Fig. 4d, lanes 7–10). This further confirms that Prd1 is a bona fide matrix protease. We speculated about the functional role of Prd1 in the mitochondrial matrix. It has been suggested that Prd1 might degrade mitochondrial presequence peptides in the IMS[39]. Therefore, we now assessed peptidase activity of Prd1 for a genuine matrix-generated presequence peptide (Cox4) which is processed by the mitochondrial presequence protease MPP upon import of precursor proteins[40]. We generated cell-free translated Prd1 protein and found a clear degradation activity for this presequence peptide compared to a mock translation (control) and a Prd1 variant with a point mutation in the catalytic site (Prd1$^{E502Q}$) (Fig. 4e, lanes 1–12). Furthermore, the full-length preprotein of Cox4 was not degraded (Fig. 4e, lanes 13–25).

Our results solve reported conflicts on the sublocalization of several proteins and uncover with Prd1 a presequence peptide degrading enzyme in the matrix. This emphasizes the high validity and quality of our landscape of submitochondrial protein distribution (Fig. 4f).

## Discussion
So far, no study has resolved mitochondrial proteins distribution to the four subcompartments on a global scale. Whereas powerful methods for the system-wide study of subcellular protein

localization have been developed[14, 41–44], no such strategy was available for deciphering suborganellar distribution of soluble and membrane proteins within double membrane-enclosed mitochondria. Indeed, a myriad of mitochondrial proteins were either not annotated to a specific subcompartment or simply mis-annotated (due to the various reasons outlined above), underlining the necessity of an unbiased approach to classify the submitochondrial proteome under exactly defined experimental conditions.

Our approach to integrate basic profiling of mitochondrial protein characteristics (solubility, membrane association or integration) with various enrichment strategies (OM vs. IM and total yeast vs. mitochondria) and the high quality reference data of the smallest subproteomes (OM and IMS) led to a landscape that captures the distribution of authentic, non-tagged proteins within the mitochondrial subcompartments. The quality of this landscape is demonstrated by solving several existing conflicts concerning sublocalization data annotated in the common databases, resulting in the reassignment of mitochondrial proteins to their actual subcompartment. Our landscape also provides a more accurate assignment of proteins to their respective organellar subcompartment compared to previous approaches, which employed spatially restricted enzymatic tagging followed by analysis with MS[14, 15]. Such strategies allow the identification of proteins in the spatial proximity of a tagged bait protein but cannot differentiate from which exact subcompartment these originate. Therefore, these approaches are more suited for determining the toponomes of cellular compartments (e.g., membrane proteins with loops reaching a soluble compartment) rather than accurately identifying the entire set of proteins within a membrane or a membrane-enclosed soluble compartment.

Yeast is a powerful model organism, which allowed the discovery of many fundamental cellular functions such as cell cycle regulation, protein sorting or autophagy. Many crucial mitochondrial functions, e.g. the protein import machineries, principles in bioenergetics, metabolite transport, interorganellar communication by the mitochondrial contact site and cristae organizing system MICOS, were first identified in this unicellular eukaryote[45–50].

The landscape of submitochondrial protein distribution elaborated here provides a resource that should foster further discoveries in this fascinating and essential organelle.

## Methods

**Yeast strains and growth conditions**. The *Saccharomyces cerevisiae* strains used in this study are derived from YPH499 (*Mat*a, *ade2-101*, *his3-Δ200*, *leu2-Δ1*, *ura3-52*, *trp1-Δ63*, and *lys2-801*[51]. Cells were grown at 24 °C on non-fermentable YPG medium (1% w/v yeast extract, 2% w/v bacto peptone, and 3% w/v glycerol (pH 5.0)). For SILAC-analysis *arg4Δ* cells[17] were grown on minimal medium (0.67% w/v yeast nitrogen base without amino acids, 3% w/v glycerol, 0.77% w/v Complete Supplement Mixture minus lysine and arginine). The medium was supplemented with natural arginine and lysine (light) or [$^{13}C_6$/$^{15}N_4$]arginine and [$^{13}C_6$/$^{15}N_2$] lysine (heavy).

**Isolation and purification of mitochondria**. Yeast cells were grown in YPG or minimal medium to an optical density (OD$_{600}$) of 0.7–1.0 (for the preparation of highly purified mitochondria and OM vesicles). For in organello import experiments cells were harvested at an OD$_{600}$ of 0.7–1.5 Cells were pelleted and resuspended in 7 ml g$^{-1}$ wet weight Zymolyase buffer (1.2 M sorbitol, 20 mM potassium phosphate-HCl (pH 7.4) containing 3 mg g$^{-1}$ wet weight Zymolyase-20T (Seikagaku Kogyo, Tokyo, Japan). After incubation for 30 min at 24 °C spheroplasts were homogenized with a glass-Teflon potter (20 strokes on ice) in homogenization buffer (0.6 M sorbitol, 10 mM Tris-HCl (pH 7.4), 1 mM EDTA, 1 mM PMSF, 0.2% (w/v) bovine serum albumin (fatty acid-free, Sigma)). Crude mitochondrial fractions were obtained from the pellet after centrifugation of the homogenate at 12,000×g for 15 min in the presence of 1x protease inhibitor cocktail (Roche)[16, 52]. Aliquots were stored in SEM buffer (250 mM sucrose, 1 mM EDTA, 10 mM MOPS-KOH (pH 7.2)) at −80 °C. Mitochondrial fractions were loaded onto a

three-step sucrose gradient (1.5 ml 60%, 4 ml 32%, 1.5 ml 23%, and 1.5 ml 15% sucrose in EM buffer (1 mM EDTA, 10 mM MOPS-KOH (pH 7.2)) to obtain highly pure mitochondria. The samples were centrifuged for 1 h at 134,000×g and highly pure mitochondria were recovered from the 32–23% sucrose interface[16, 52]. Mitochondrial purity was assessed by western blotting against various cellular marker proteins.

To obtain the post-mitochondrial fractions S100 and P100 homogenized yeast cells were centrifuged at 20,000×g for 10 min at 4 °C. The supernatant was subjected to two further identical centrifugation steps followed by centrifugation at 100,000×g for 1 h resulting in the supernatant (S100, containing mainly cytosolic proteins) and the pellet (P100, containing largely microsomal fraction). The pellet was resuspended in EM buffer and fractions were analyzed via SDS–PAGE and immunoblotting.

**Generation of submitochondrial fractions**. For sonication, 50–500 μg mitochondria were suspended in 1 mL SEM buffer in the presence of 500 mM NaCl (or 50 mM for OM$_L$/TOT$_H$ samples). Samples were sonicated for $3 \times 30$ s on ice and subsequently centrifuged at 4 °C for 1 h at 100,000×g. For carbonate extraction, 50–500 μg mitochondrial proteins were incubated in 400–1000 μL freshly prepared 100 mM sodium carbonate. After incubation on ice for 30 min with occasional vortexing, an ultracentrifugation step was performed at 100,000 g for 45 min at 4 °C. Supernatants of both treatments were precipitated with 10% w/v trichloroacetic acid. Proteins were subjected to LC-MS/MS analysis or solubilized in Lämmli buffer, separated by SDS–PAGE and analyzed by Western blotting.

For swelling, mitochondria were suspended in 400 μL EM buffer and incubated on ice for 30 min with occasional mixing followed by centrifugation for 15 min at 20,000 × g. Obtained supernatants were precipitated with 10% w/v trichloroacetic acid. Pellet and supernatant samples were analyzed by SDS–PAGE and immunodecoration.

Release of soluble mitochondrial proteins by Bax treatment was carried out as previously described[17]. Mitochondria were incubated for 1 h at 37 °C in buffer consisting of 250 mM sucrose, 150 mM KCl, and 10 mM MOPS-KOH (pH 7.2) in the presence or absence of 100 nM human Bax. Supernatant and pellet fractions were separated by centrifugation at 16,000×g for 15 min at 4 °C.

For the isolation of OM vesicles, highly pure mitochondria were swollen in hypoosmotic buffer (5 mM potassium phosphate (pH 7.4), 1 mM PMSF) at a concentration of 4 mg ml$^{-1}$. After incubation on ice for 20 min sample was treated with a glass-Teflon potter (20 strokes). The homogenate was then subjected to two consecutive ultracentrifugation steps on discontinuous sucrose gradients as described[31] to recover OM vesicles.

**Protease activity assay**. Cell-free translated Prd1 and Prd1$^{E502Q}$ (point mutation by site-directed mutagenesis) was generated using the RTS100 wheat germ system (5PRIME). Prd1 was incubated with 15 μM Cox4 presequence peptide (MLSLRQSIRFFKPATRT) or radiolabelled Cox4 precursor protein in import buffer without BSA supplemented with 1× Complete, EDTA-free protease inhibitor cocktail (Roche)[40]. Samples were separated on Nu-PAGE (Novex) and subjected to immunoblotting.

**Sample preparation for mass spectrometry**. In total four replicates (two biological replicates with each time one forward (e.g., SN$_{heavy}$/PEL$_{light}$) and one reverse experiment (e.g., SN$_{light}$/PEL$_{heavy}$) respectively; Fig. 1a) were used. Material was isolated from four independent yeast cultures, of which two were grown with supplementation of heavy and two with supplementation of light amino acids (see Yeast strains and growth conditions for details). Sample amounts were equalized based on Bradford protein determination. Highly purified mitochondria were generated and subjected to sonication or carbonate extraction. Next, heavy and light samples were pooled 1:1 to determine the following ratios for four independent replicates: (i) supernatant sonication (SN$_{son}$) vs. pellet sonication (PEL$_{son}$), (ii) supernatant carbonate (SN$_{carb}$) vs. pellet carbonate (PEL$_{carb}$). Cysteines were reduced with 10 mM DTT for 30 min at 56 °C, and free sulfhydryl groups were carbamidomethylated using 30 mM iodoacetamide for 30 min at room temperature in the dark. Proteins were precipitated with a 10-fold excess of ethanol for 1 h at −40 °C, followed by centrifugation at 14,000×g at 4 °C for 30 min. Obtained pellets were washed with acetone, followed by centrifugation as above for 15 min. Samples were resuspended in 2 M GuHCl, 50 mM NaH$_2$PO$_4$ (pH 7.8), and diluted 10-fold with 50 mM NH$_4$HCO$_3$. Acetonitrile (ACN) and CaCl$_2$ were added to final concentrations of 5% and 1 mM, respectively. Trypsin (Promega, sequencing grade) was added at a ratio of 1:30 and incubated at 37 °C for 12 h. Peptide samples were desalted using SPEC C18 AR tips (Agilent, Waldbronn, Germany) according to the manufacturer's instructions, and dried under vacuum. Samples were resuspended in 10 mM KH$_2$PO$_4$ (pH 2.7) and fractionated using strong cation exchange chromatography (SCX[17]). Digests were controlled by monolithic column HPLC[53].

**Strong cation exchange chromatography**. SCX was performed using a self-packed 150 mm × 550 μm PolySULFOETHYL A column (200 Å pore size,

5 μm particle size; PolyLC, Columbia, MD, USA) in combination with an Ultimate 3000 HPLC system (Thermo Fisher, Germering, Germany). Peptides were separated at a flow rate of 20 μL min$^{-1}$ using a binary gradient (SCX buffer A: 5 mM KH$_2$PO$_4$ (pH 2.7) 20% ACN (pH 2.7); SCX buffer B: 5 mM KH$_2$PO$_4$, 200 mM KCl, 20% ACN (pH 2.7)) ranging from 0 to 95% B in 50 min. Six fractions were collected in concatenation mode, as previously described[17]. Fractions were desalted using self-packed Oligo R3 (Thermo Scientific) microcolumns, dried under vacuum and resuspended in 0.1% trifluoroacetic acid (TFA).

**Nano-LC-MS/MS analysis.** Nano-LC-MS/MS was performed on an LTQ-Orbitrap Elite mass spectrometer coupled to an Ultimate 3000 RSLC (both Thermo Fisher Scientific). Briefly, peptides were preconcentrated on a C18 trapping column (Acclaim PepMap, 100 μm × 2 cm, 5 μm particle size, 100 Å pore size, Thermo Fisher Scientific) in 0.1% TFA and separated on a C18 main column (Acclaim PepMap, 75 μm × 50 cm, 2 μm particle size, 100 Å pore size, Thermo Fisher Scientific) using a binary gradient (solvent A: 0.1% formic acid (FA); solvent B: 0.1% FA, 84% ACN) ranging from 3 to 42% B in 185 min, at a flow rate of 250 nL min$^{-1}$. MS survey scans were acquired in the Orbitrap from 300 to 2000 $m/z$ at a resolution of 60,000 using the polysiloxane $m/z$ 371.1012 as a lock mass. The 20 most intense signals above an intensity threshold of 10$^4$ and with charge states 2–5 were subjected to collision induced dissociation in the ion trap with a normalized collision energy of 35%, taking into account a dynamic exclusion of 10 s. Automatic gain control (AGC) target values and maximum injection times were set to 10$^6$ and 50 ms for MS and 10$^4$ and 100 ms for MS[2].

**Data interpretation and protein assignments.** Data interpretation was conducted with the help of MaxQuant (v 1.305) using Andromeda and the Saccharomyces Genome Database (February 2011; 6750 target sequences) and the following settings: (i) trypsin without missed cleavages; (ii) carbamidomethylation of cysteine as fixed and (iii) oxidation of methionine, acetylation of protein N-termini, $^{13}C_6^{15}N_2$ Lys and $^{13}C_6^{15}N_4$ Arg as variable modifications; and (iv) MS and MS/MS tolerances of 10 ppm and 0.5 Da, respectively. Only unique peptides were considered for quantification at a false discovery rate of <1% (peptides and proteins). The eight data sets were merged to a master table and only proteins for which at least two unique peptides were quantified for the sonication or the carbonate data sets were considered to determine average ratios from the four SN sonication vs. PEL sonication ratios (SN$_{son}$/PEL$_{son}$) and the four SN carbonate vs. PEL carbonate ratios (SN$_{carb}$/PEL$_{carb}$). The reference proteomes of resident proteins of the OM and IMS were extracted from ref. [31] including three novel OM proteins Mim2, Caf4 and Atg32, and ref. [17], respectively (Supplementary Data 5). For the reference set of Supplementary Fig. 2 each protein was manually reviewed using the original literature containing biochemical data on their sublocalization. The most frequent contaminant in mitochondrial proteomic studies, Plasma membrane ATPases PMA1 and PMA2[6, 31] were not further assigned in this study. Novel mitochondrial proteins were identified by their absence in both, the PROMITO data set[6, 7] and SGD (manually curated annotations; Version July 2016)[8].

**Statistical model and prediction.** We set up a statistical model describing the distribution of proteins. We used our reference set of proteins with well-known localization (Supplementary Fig. 2) as protein list $P_L$ whereas all others represented list $P_U$. We created a model using solely $P_L$ and applied this model on $P_U$. Therefore, we used the ratios $s = \log_{10}(SN_{son}/PEL_{son})$ and $c = \log_{10}(SN_{carb}/PEL_{carb})$ for sonication (S) and carbonate extraction (C). The two-dimensional multivariate normal distribution was taken as the model with parameters

$$\mu = (\mu_S, \mu_C), \ \Sigma = \begin{pmatrix} \sigma_S^2 & \rho\sigma_S\sigma_C \\ \rho\sigma_S\sigma_C & \sigma_C^2 \end{pmatrix}$$

where $\rho$ is the correlation between S and C. The probability density function is defined as

$$f(s, c \mid \mu, \Sigma) = \frac{1}{2\pi\sigma_S\sigma_C\sqrt{1-\rho^2}} \exp\left(\frac{-1}{2(1-\rho^2)}\left[\frac{(s-\mu_S)^2}{\sigma_S^2} + \frac{(c-\mu_C)^2}{\sigma_C^2} - \frac{2\rho(s-\mu_S)(c-\mu_C)}{\sigma_S\sigma_C}\right]\right).$$

To estimate the parameters, we utilized the maximum-likelihood estimators, that is

$$\hat{\mu} = \frac{1}{|P_L|}\sum(s,c), \hat{\Sigma} = \frac{1}{|P_L|}\sum((s,c)-\hat{\mu})\cdot((s,c)-\hat{\mu})^T$$

We obtained the following parameters for the three stages:

$$\text{Soluble}: \hat{\mu}(1.0273, 1.2561), \hat{\Sigma}\begin{pmatrix} 0.0640 & 0.0054 \\ 0.0054 & 0.0289 \end{pmatrix};$$

$$\text{Peripheral}: \hat{\mu}(-0.0351, 1.1155), \hat{\Sigma}\begin{pmatrix} 0.1355 & 0.0649 \\ 0.0649 & 0.0797 \end{pmatrix};$$

$$\text{Integral}: \hat{\mu}(-0.5041, -0.6293), \hat{\Sigma}\begin{pmatrix} 0.0373 & 0.0179 \\ 0.0179 & 0.2459 \end{pmatrix}.$$

To determine the probability $p$ that a protein of $P_U$ belongs to a certain state, we computed the average over all three states adding an equally distributed background model $b$ for proteins that cannot be described by any state, let

$$p_{i,\text{soluble}} = \frac{f((s,c)_i|\mu_{\text{soluble}}, \Sigma_{\text{soluble}})}{f((s,c)_i\mu_{\text{soluble}}, \Sigma_{\text{soluble}}) + f((s,c)_i|\mu_{\text{peripheral}}, \Sigma_{\text{peripheral}}) + f((s,c)_i|\mu_{\text{integral}}, \Sigma_{\text{integral}}) + b},$$

$$p_{i,\text{peripheral}} = \frac{f((s,c)_i|\mu_{\text{peripheral}}, \Sigma_{\text{peripheral}})}{f((s,c)_i|\mu_{\text{soluble}}, \Sigma_{\text{soluble}}) + f((s,c)_i|\mu_{\text{peripheral}}, \Sigma_{\text{peripheral}}) + f((s,c)_i|\mu_{\text{integral}}, \Sigma_{\text{integral}}) + b},$$

$$p_{i,\text{integral}} = \frac{f((s,c)_i|\mu_{\text{integral}}, \Sigma_{\text{integral}})}{f((s,c)_i|\mu_{\text{soluble}}, \Sigma_{\text{soluble}}) + f((s,c)_i|\mu_{\text{peripheral}}, \Sigma_{\text{peripheral}}) + f((s,c)_i|\mu_{\text{integral}}, \Sigma_{\text{integral}}) + b}$$

for every $1 \le i \le |P_U|$.

Here, we set $b = (0.5 \cdot (\max(s) - \min(s))(\max(c) - \min(c)))^{-1} = 19.482$.

Since we assumed that all proteins are equally distributed among all three states, we did not weight the three models. In case a certain state described a protein with a probability >0.5, the protein was discretely assigned to this state. The same holds true in case the probability was 0.4–0.5 by model A (without loss of generality) while concurrently the probability by model B was at least 0.15 smaller. Proteins passing neither criteria were considered as ambiguous. Thus, the model confirmed 94.1% of the original assignments with clear $s$ and $c$ ratios (200/214 soluble, 253/255 peripheral, 292/302 integral, and 152/182 ambiguous). Supplementary Fig. 3 represents the distribution of the reference set $P_L$, whereas Supplementary Fig. 4 illustrates the distribution of all proteins $P_L + P_U$. The color coding is red = peripheral, green = integral, and soluble = blue. For every protein, all three color channels RGB were multiplied by their corresponding probabilities to indicate memberships. The model boundaries indicate 80% of each density.

**Comparison of total yeast vs. total mitochondrial proteomes.** To assess the likelihood that new mitochondrial proteins are also located in other cellular compartments or might represent contaminants, we quantified the complete yeast and mitochondrial proteomes derived from *arg4Δ* cells grown under non-fermentative conditions (see above). Yeast cells and highly pure mitochondria were lysed in 10% w/v SDS, 150 mM NaCl, 50 mM Tris (pH 8.0). The protein content was determined using the bicinchoninic acid assay (BCA, Thermo Scientific) according to the manufacturer's instructions. Proteins were carbamidomethylated, precipitated, digested, and samples desalted as described above. 50 μg of digest (mitochondria and yeast) were resuspended in 10 mM ammonium acetate (pH 6.5) and fractionated on a C18 column (Zorbax, I.D. 0.5 mm × 150 mm, Agilent) at pH 6.5 (solvent A: 10 mM ammonium acetate (pH 6.5); solvent B: 10 mM ammonium acetate, 84% ACN) using a 90 min gradient ranging from 3 to 50% B. 20 fractions were collected in a concatenated manner, dried under vacuum, resolubilized in 0.1% TFA. Per fraction 50% was analyzed by nano-LC-MS/MS on Q Exactive Plus coupled to a U3000 RSLC system (both Thermo Fisher Scientific) using LC parameters as above (2 h gradient). MS scans were acquired at a resolution of 70,000, the 15 most abundant ions were fragmented by higher energy collisional dissociation with a normalized collision energy 30% and MS/MS were acquired at a resolution of 17,500. AGC and maximum injection times were set to 3 × 10$^6$ and 120 ms for MS and 2 × 10$^5$ and 250 ms for MS/MS scans. Raw data were converted into mascot generic files using Proteowizard (Version 2.2.2954) and searched with Mascot, OMSSA and X!Tandem using SearchGUI 1.12.2[54]. Data were analyzed using PeptideShaker version 0.20.1[55] at a false discovery rate of 1% on the protein, peptide and peptide-spectrum-match (PSM) levels. For the mitochondrial proteome, normalized spectral abundance factors (NSAF)[56] were calculated for estimating the relative amounts of all identified proteins across the mitochondrial

proteome. To assess, whether new candidate proteins are located exclusively in mitochondria or also in other cellular compartments, we furthermore calculated NSAF of these proteins in the whole yeast sample, i.e., only considering PSM of mitochondrial proteins. Per protein a reference yeast/mito ratio was calculated by dividing the corresponding yeast NSAF by the mitochondrial NSAF. Thus, proteins that have dual (multiple) subcellular localizations or represent potential contamination should have high yeast/mito ratios, corresponding to a relative enrichment in the yeast proteome as compared to the mitochondrial proteome.

**Identification of integral inner and outer membrane proteins.** To specifically distinguish integral membrane proteins from the inner and outer mitochondrial membranes, we compared the proteomes of carbonate extracted (i) highly purified OM vesicles ($OM_{Light}$) and (ii) total membranes ($TOT_{Heavy}$), containing both organellar membranes. Carbonate resistant pellets from $OM_{Light}$ and $TOT_{Heavy}$ samples, according to BCA both ~3 μg, were ultracentrifuged in SEM buffer at 100,000×$g$ and 4 °C for 1 h. Pellets were resolubilized in 2 M GuHCl, 50 mM $NaH_2PO_4$ (pH 7.8) and processed as described above. After digestion, samples were pooled in three different proportions: 1 μL $OM_{Light}$ + 10 μL $TOT_{Heavy}$ (1:10), 2:10 and 3:10. Pooled samples were analyzed on a Q Exactive Plus as described above. Data analysis was conducted using MaxQuant[57] as described above. As several proteins were absent in either light/heavy samples, $OM_L$ and $TOT_H$ SILAC intensities were used to calculate per protein its relative intensity in the OM fraction. Values of all three samples were used to determine the average relative intensities in the OM fraction. For integration of the data into the landscape only identified proteins from this study and resident OM proteins of the OM reference proteome (see above) were considered.

**In organello import of mitochondrial precursor proteins.** Radiolabeled precursors were generated in vitro with the transcription and translation rabbit reticulocyte lysate system (Promega) supplemented with [$^{35}$S]methionine. Mitochondria (80 μg) and precursor proteins were incubated at 30 °C for 45 min in import buffer (10 mM MOPS-KOH (pH 7.2), 3% w/v bovine serum albumin, 250 mM sucrose, 5 mM $MgCl_2$, 80 mM KCl, and 5 mM $KP_i$). Samples were supplied with 2 mM ATP and 2 mM NADH. Import reactions were abolished or terminated by disruption of the membrane potential by addition of 8 μM antimycin A, 1 μM valinomycin, and 20 μM oligomycin (AVO). Non-imported precursors were digested with 50 μg ml$^{-1}$ Proteinase K or 25 μg ml$^{-1}$ Trypsin and incubation for 10–15 min on ice. Mitochondria were reisolated by centrifugation at 16,000×$g$ for 10 min at 4 °C and washed with SEM buffer. Analysis was performed by SDS–PAGE and digital autoradiography using the FLA 9000 image scanner (Fujifilm) and the freeware ImageJ version 1.40 g (National Institutes of Health).

**Miscellaneous.** Antibodies were generated by immunization of rabbits using synthetic peptides (Supplementary Table 1). Antibodies against the candidate proteins with multiple localizations were raised against the following peptides: Lap3, KEEPIVLPIWDPMGALAK; Gsf2, GEDLKKFRKIRKEQDPDN; Ape4, FKEFFERYTSIESEIVV; Ape2, NRDRDVVNKYLKENGYY; Lsp1, ADHHVSQNGHTSGS ENI; and Zeo1, EKKETKKEGGFLKKLNRK. Peptides were coupled to keyhole limpet hemocyanin via N-terminal cysteines. Western blotting was performed according to standard protocols.

**Data processing.** Photoshop CS5 (Adobe) was used to process images and the figures were compiled using Illustrator CS5 (Adobe). To show regions of interest blots and autoradiography scans were digitally processed. Uncropped versions of immunoblots and autoradiographs are shown in Supplementary Figs. 8–10.

**Data availability.** The mass spectrometry proteomics data have been deposited to the ProteomeXchange Consortium via the PRIDE partner repository[58] with the data set identifiers PXD005463 and PXD005541. Further relevant data can be obtained from the authors upon request.

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

## Acknowledgements

We thank B. Schönfisch, C. Prinz and L. Myketin for expert technical assistance. We thank Dr. J.C. Martinou for the Bax protein. Work included in this study has also been performed in partial fulfillment of the requirements for the doctoral thesis of C.K. This work was supported by the Deutsche Forschungsgemeinschaft, Excellence Initiative of the German Federal & State Governments (EXC 294 BIOSS), the RTG GRK2202 (to C.M.), the Ministerium für Innovation, Wissenschaft und Forschung des Landes Nordrhein-Westfalen (to R.P.Z., J.M.B., and H.G.J.), the Emmy-Noether Programm of the Deutsche Forschungsgemeinschaft to F.N.V. and the CAPES Foundation to H.G.J.

## Author contributions

F.-N.V., J.M.B., H.G.-J., C.K., A.A.T., and D.M. performed the experiments. D.K. and R. A. performed statistical analysis. F.-N.V., J.M.B., R.P.Z., and C.M. designed experiments, analyzed, and interpreted the data. C.M., F.-N.V., J.M.B., and R.P.Z. developed the project and wrote the manuscript. H.G.-J. and A.S. reviewed and edited the manuscript. C.M. and R.P.Z. coordinated and directed the project. All authors approved the final version of the manuscript.

## Additional information

**Competing interests:** The authors declare no competing financial interests.

