## [Peer Review File · Nature Communications]

Reviewers' Comments:

Reviewer #1 (Remarks to the Author):

In this manuscript Vogtle and colleagues define the sublocalization of mitochondrial proteins on a proteomic basis and provide a comprehensive view of protein distribution across four mitochondrial subcompartments. In the first part of this proteomic study the authors biochemically separate proteins based on their ability to be extracted from the membranes via carbonate treatment. Combining this method with a MS-SILAC approach, they classify proteins into three different clusters: integral, soluble and peripheral membrane proteins. This analysis also identifies novel mitochondrial proteins that have not been annotated so far. In the second part of the manuscript the authors define for the first time a compendium of the integral inner membrane proteins utilizing the SILAC ratio of outer membrane proteins to the total mitochondrial membrane proteins. Integrating both analyses the authors generate a master map of the submitochondrial protein distribution.

This is an excellent study. The experiments presented in this manuscript are rigorously done, clearly presented and multiple approaches were used to validate the results. Besides its technical quality, the study has high significance and provides unprecedented overview on submitochondrial protein localization. The fact, that databases contain a large amount of wrong information, which stems from limited experimental data, I find that this is a very important contribution to the field of molecular cell biology, which will hopefully be used to fix the database shortcomings. The study will be of broad interest and fully suitable for publication in Nature communications.

I have only a few minor points:

1. In Fig. 4e Vogtle et al. determine the peptidase activity of Prd1 claiming this protein is a presequence peptide degrading enzyme. However, in the assay they don't include negative controls (e.g. not degradable protein). Such a control would be required.
2. It would strengthen the manuscript, if the authors would include a carbonate extraction analysis of some of the dually localized proteins that are clustered in the ambiguous region (Suppl. Fig. 3).
3. In Suppl. Fig. 2 the authors display clusters of three different classes of proteins. Few of them

belong to the integral membrane cluster, e.g. Mba1 or Atp7 are well known peripheral proteins. The authors should comment on this in the text.

4. On page 3, 17 and 20 the authors state that the SILAC ratios were determined from four biological replicates, although it seems from the Supplemental Table 1 that there are only 2 data sets. It would be good to clarify in the text that the SILAC ratio were defined from 2 biological replicates: 2 forward experiments (e.g. SNheavy/Pelletlight) and 2 reverse experiments (e.g. SNlight/Pelletheavy) for each condition, sonication and carbonate.

5. In Figure 3b the authors analyzed the subcellular distribution of selected proteins by Western analyses. Ape4 in Mito. and S100 fractions migrate differently. Is there an explanation for this? Lsp1 runs as a double band. Only in the P100 it appears to be solely the faster migrating form. Is this protein processed?

6. Typos:

e.g. page 7, ref. 29 (dot); page 7, protein8 (superscript);

page 5, Suppl. Table 2 (instead of S2);

legend of the Supplementary Figure 2 (instead of S2).

7. On Page 3 the authors use the term “SILAC-labelled “ this appears to me as a tautology as SILAC stands for Stable Isotope Labelling in Cell Culture. This could be easily fixed.

8. Some entries in the list of references are not formatted correctly:

e.g. page 11, reference 11: et al. (Italic);

page 13, reference 34: et al. (Italic); page 13, reference 44: 55 (Bold);

page 25, reference 54: 44 (Bold).

Reviewer #2 (Remarks to the Author):

Review of Vögtle et al.

This paper concerns the mapping of proteins to sub-mitochondrial compartments, using organelle enrichment followed by submitochondrial fractionation on heavy and light SILAC labelled yeast cultures. The authors use well-established methods to sub-fractionate the mitochondria, including carbonate washing and sonication to obtain soluble protein and integral/peripheral membrane fractions. This results in a number of scores, which the authors use to estimate the enrichment of proteins in each mitochondrial sub-compartment (integral membrane, peripheral membrane or soluble) and leads to a map of the organelle in which different sub-compartment resolve from

each other and form distinct clusters, with multiply localised proteins contained in the intervening space. The use of the ratios to estimate enrichment and infer sub-mitochondrial localisation of proteins is quite a nice approach to take. The authors also carry out comparison using label-free proteomics (spectral counts) to demonstrate enrichment of candidate novel mitochondrial proteins from their initial map, in the mitochondrial fraction relative to a whole-cell extract. This gives also associated ratios from which one can estimate the relative enrichment of a protein in the mitochondrial fraction and infer its localisation.

The authors demonstrate enrichment of some of their predicted novel mitochondrially localised proteins, using crude differential centrifugation (crudely separating microsomes, mitochondria and cytosol), which shows enrichments of mitochondrial proteins as they expect and lack of enrichment of non-mitochondrial proteins. They further demonstrate import of some of their putative novel mitochondrial proteins from their SILAC label-based sub-mitochondrial map (integral membrane vs. soluble vs. peripheral membrane) using radioactive in organello import assay. They select a subset of proteins from either the peripheral, soluble or ambiguous regions of their map, allow import and perform mitochondrial sub-fractionation by a protease protection assay. This is followed by SDS-PAGE and autoradiography, to demonstrate whether radioactive proteins are been imported. A subset (12 proteins) of their predicted novel 206 mitochondrial proteins are biochemically validated using this method, (3 predicted ambiguous, 5 predicted soluble, and 4 predicted peripheral membrane proteins) (page 6 of the article). This represents less than 10% of the putative proteins and it would have been good to see additional proteins validated in this manner. Further, might it be appropriate to validate some of these observations in their system using a visual method such as super-resolution microscopy (e.g. STED) or immunogold staining (Gripic et al., 2004; Wolff et al., 2014), searching for co-localisation of the putative mitochondrially-localised proteins with some of the proteins from the authors' reference set of mitochondrially-localised proteins. Additionally the authors could consider determining protein localisation in their system using strains from the Yeast GFP clone collection as has been performed previously (Breker et al., 2013; Chong et al., 2015; Déneraud et al., 2013; Huh et al., 2003), rather than relying on this import assay. This would offer more information regarding potential multiple localisations than can be observed from the import assay or differential centrifugation, and would even directly show visually that the protein is localised in the mitochondrion.

The authors also undertake to distinguish between integral membrane proteins of the outer membrane and inner membrane using highly purified outer membrane vesicles from a light SILAC labelled culture and total membranes isolated by sodium carbonate wash from a heavy SILAC labelled culture. They work out SILAC ratios to characterise enrichment in the outer membrane fraction relative to total membranes and infer the presence and thus localisation of integral membranes from either the inner membrane or the outer membrane. This analysis must be contingent on obtaining highly pure outer membranes. The method used to obtain such outer

membrane vesicles is from a paper published in 2006 by one of the authors (their reference 29). The characterisation of the outer membranes in that paper was performed by excision of spots from a 2D-PAGE gel of isolated outer membrane and characterisation of the spots using mass spectrometry. There is no validation or demonstration as to the purity or enrichment of their outer membranes in the current article. I think that this should be demonstrated by some method (e.g. utilising immunoblotting, probing with a panel of antibodies raised against integral inner or outer membrane proteins) to demonstrate such enrichment, before any conclusions are drawn as to the localisation of any protein to either mitochondrial membrane.

In general I think that the results are impressive and show good clustering of the sub-mitochondrial proteins (in Supplementary Figures 1 and 2), but the methods used to lyse and fractionate the cells, and the authors' lack of demonstration of some of the data, are somewhat concerning. Firstly, the method used to lyse the cells is not stated, and the results of this study are highly contingent on the presence of highly enriched mitochondria which should be intact until such time as they are sub-fractionated. Further, the authors' statement that they "highly purified" their mitochondria and confirmed purity by western blotting is concerning as the western blots are not shown in this manuscript.

How do we know that the mitochondria are pure or enriched to a high enough degree? How do we know whether the mitochondria are intact and not damaged? Further their methods section seems to suggest that they isolate only the interphase of the density gradient used for mitochondrial purification. They assume this interphase to contain the highly pure mitochondria, but nothing else. How can they be sure that there is not more than one population of mitochondria unless they sample protein from all parts of their density gradient? It is also somewhat concerning that the cells are lysed and then frozen at -80°C before thawing and performing cellular fractionation. Freeze-thaw is damaging to proteins (Cao et al., 2003) and often used to lyse cells suggesting that it can be damaging to organelles and subcellular structures. This supports further that the authors should more thoroughly analyse their whole density gradient, and demonstrate this analysis (by western blotting or other means), before they make any assumptions as to the purity of their mitochondrial preparation and draw any conclusions from it.

The authors construct a reference set of mitochondrial proteins, which they use to demonstrate separation achieved by their sub-fractionation approach. This reference set is plotted on the map in Supplementary Figure 1, and represents members of each of the sub-organellar niches. The authors already say, however, in their introduction that the assignment of proteins to mitochondria is problematic and a lot of the annotation to the organelle is based on data which they do not trust. How exactly was this reference set constructed? What were the exact criteria?

In Supplementary Figures 1, 2 and 3 (the mitochondrial map figures), how are the ellipses

representing the clusters constructed? They seem slightly arbitrary as there do not seem to be bounds for the scores used to define their clouds. What governed the scores which were used to define membership of a specific part of the mitochondrion? How can they therefore be used to ascertain whether the proteins contained within are part of these sub-structures? How do they know that their clouds encompass the entirety of the specific sub-structure? How do they know that the clouds are elliptical and not some other shape unless they apply some sort of confidence metric or bounds to these data? I would have expected some statistical tests to be applied to determine the boundaries of the clusters and some estimate of the false discovery rates of assignment to the resulting clusters. When the authors have assigned some sort of confidence metric to their assignments, it may be useful to validate some of their specific localisation predictions. This would be performed on a subset of their high and low confidence novel proteins which are predicted to localise to each of the specific sub-mitochondrial clusters, and could include validation by other means such as super-resolution microscopy.

The authors observe in the “integral membrane” fraction in each of their mitochondrial maps, the yeast plasma membrane ATPases Pma1p and Pma2p. They describe these proteins as the “most frequent contaminants in proteomics studies” citing their own papers, in which these proteins were contaminants in mitochondrial preparations. In these papers the reason given for contamination by these proteins is that they are easily accessible to tryptic digestion. In one such paper, this statement is backed up with an observation from the work of Washburn et al. (Washburn et al., 2001), which itself does not in fact say that these proteins are contaminants in proteomics studies. I therefore do not think that this is a particularly strong argument. The authors say that the supposed accessibility of Pma1p and Pma2p to tryptic digest enables these proteins to be detected even in “tiny amounts”. The authors thus discount them from further analysis. If this were the case, would it not be expected to be true for other proteins which exhibit equal accessibility to tryptic digest? The presence of these proteins in their preparations could be indicative of the purity or degree of enrichment of their mitochondrial samples and cast doubt on some of their other observations regarding “novel” mitochondrial protein assignments. To make such a statement, it is argued that they should find evidence for this observation from other sources.

In summary, the manuscript describes a huge amount of work to determine the submitochondrial proteome and the data presented have the potential to be of high utility to researchers. As currently presented the study falls short in two main areas; demonstration of the purity of mitochondria achieved and their integrity after purification/enrichment, robust statistical analysis of the resulting data. Without these shortcomings being addressed, the conclusions made in this study cannot be fully supported and hence the manuscript is not yet ready for publication in Nature Communications.

Specific smaller comments:

The order of the introduction seems a little odd with the results summarised before being put into context, i.e. previous proteome maps of the mitochondrial being far from complete or well resolved into subcompartments.

In the introduction (page 1), the authors say that 986 proteins were assigned, but do not say whether they were assigned to a specific sub-organellar location or to the organelle in general. This should be made clearer.

On page 3 (results) they say that they isolated highly pure light and heavy mitochondria, but it is unclear from this text whether they mean light membranes and heavy membranes or light SILAC labelled and heavy SILAC labelled mitochondria. This should be made clearer.

On page 4, they say that a subset of proteins in their map are localised to the “correct” clusters, but they do not know that their clusters are correct. “Expected” would be better to use here, as in the absence of further validation they do not know whether their assumptions are correct.

They also make a sweeping statement that the mitochondrial proteome consists of the proteins that they have detected in their study. This is a bit of a generalisation, as they could say that “their” mitochondrial proteome consists of these proteins. They do not know whether their proteome is exhaustive, what the level of contamination is or whether they are not sampling a fraction of the mitochondrial proteome.

On page 5, they mention that the “majority” of known mitochondrial proteins show low yeast/mito ratios using their spectral counting approach, but it is not clear what this number represents as the supplementary table referenced is not available. I think that this should be changed to a definite value, even if it is available in a supplementary table.

On page 22, under Quantitative Comparison of total yeast vs. total mitochondrial proteomes, the authors state what their chromatography solvent A is, but not solvent B. This should be stated.

On page 24, the authors state that they digitally altered their western blots to remove non-relevant bands. It is argued that this is neither appropriate nor acceptable and that the scanned blots should be included in their entirety.

In their Online Methods section, the concentration of yeast nitrogen base in their minimal media used for SILAC seems to be 10 times too high at 6.7% (w/v) (normal concentration for minimal media is 0.67% (w/v)) (Sherman, 2002). Further, the OD600 to which they grow their cells has quite a wide range (0.7-1.5), which, can encompass different growth phases of yeast (from early to mid-exponential phase). It is unclear why they use such a wide range of optical densities and

this should be explained, as it might be expected that the proteins could change localisation based on growth phase.

References

- Breker, M., Gymrek, M., and Schuldiner, M. (2013). A novel single-cell screening platform reveals proteome plasticity during yeast stress responses. *J. Cell Biol.* 200, 839–850.
- Cao, E., Chen, Y., Cui, Z., and Foster, P.R. (2003). Effect of freezing and thawing rates on denaturation of proteins in aqueous solutions. *Biotechnol. Bioeng.* 82, 684–690.
- Chong, Y.T., Koh, J.L.Y., Friesen, H., Duffy, K., Cox, M.J., Moses, A., Moffat, J., Boone, C., and Andrews, B.J. (2015). Yeast proteome dynamics from single cell imaging and automated analysis. *Cell* 161, 1413–1424.
- Dénervaud, N., Becker, J., Delgado-Gonzalo, R., Damay, P., Rajkumar, A.S., Unser, M., Shore, D., Naef, F., and Maerkl, S.J. (2013). A chemostat array enables the spatio-temporal analysis of the yeast proteome. *TL - 110. Proc. Natl. Acad. Sci. U. S. A.* 110 VN-, 15842–15847.
- Griparic, L., van der Wel, N.N., Orozco, I.J., Peters, P.J., and van der Blik, A.M. (2004). Loss of the Intermembrane Space Protein Mgm1/OPA1 Induces Swelling and Localized Constrictions along the Lengths of Mitochondria. *J. Biol. Chem.* 279, 18792–18798.
- Huh, W.-K., Falvo, J. V, Gerke, L.C., Carroll, A.S., Howson, R.W., Weissman, J.S., and O’Shea, E.K. (2003). Global analysis of protein localization in budding yeast. *Nature* 425, 686–691.
- Sherman, F. (2002). Getting started with yeast. *Methods Enzymol.* 350, 3–41.
- Washburn, M.P., Wolters, D., and Yates, J.R. (2001). Large-scale analysis of the yeast proteome by multidimensional protein identification technology. *Nat. Biotechnol.* 19, 242–247.
- Wolff, N.A., Ghio, A.J., Garrick, L.M., Garrick, M.D., Zhao, L., Fenton, R.A., and Thévenod, F. (2014). Evidence for mitochondrial localization of divalent metal transporter 1 (DMT1). *FASEB J.* 28, 2134–2145.

Reviewer #3 (Remarks to the Author):

This manuscript, submitted as a Resource, attempts to generate a map of yeast submitochondrial protein localization via a series of biochemical treatments and fractionations coupled with quantitative mass spectrometry techniques. By and large, the experimental techniques are all carefully and rigorously performed and the resulting dataset is high-quality, well organized, and informative. The current version risks overreaching in certain areas and could provide additional guidance/rationale on certain parameters. However, given that the points below are addressed, this work will serve as a very useful resource for the mitochondrial community and should make for a timely Nature Communications publication.

Major points

1. In reading the introduction, one would think that considerable efforts to map sub-mitochondrial proteomes had not already been done. In particular, I find that Alice Ting's APEX2 work to be conspicuously absent. This work is later referenced in the Discussion, but it should be introduced earlier in the paper. Additionally, an effort should be made to compare the results of these efforts for proteins conserved between yeast and mammalian systems. Such analyses would be useful whether or not the data proved to be consistent.
2. The use of the yeast/mito ratio for defining novel mitochondrial proteins is underdeveloped and oversimplified as currently depicted. Of course, any co-purifying organelle fraction would also score well by this analysis. What kind of yeast/mito scores to well-established proteins from other organelles receive, and how do these compare to the novel proteins? Also, while the validation of the selected novel proteins is convincing, it is not clear why they were chosen.

Minor points

1. A short explanation should be given for a broader audience as to why some of the validated novel proteins have observable size shifts upon import and others do not
2. In the CoQ reference set and corresponding Figure 2C, it should be noted that Coq8 has a single-pass transmembrane domain. Interestingly, this seems consistent with its lower SN_{son}/PEL_{son} score compared to the other Coq-related proteins.

Reviewers' comments:

Reviewer #1 (Remarks to the Author):

In this manuscript Vogtle and colleagues define the sublocalization of mitochondrial proteins on a proteomic basis and provide a comprehensive view of protein distribution across four mitochondrial subcompartments. In the first part of this proteomic study the authors biochemically separate proteins based on their ability to be extracted from the membranes via carbonate treatment. Combining this method with a MS-SILAC approach, they classify proteins into three different clusters: integral, soluble and peripheral membrane proteins. This analysis also identifies novel mitochondrial proteins that have not been annotated so far. In the second part of the manuscript the authors define for the first time a compendium of the integral inner membrane proteins utilizing the SILAC ratio of outer membrane proteins to the total mitochondrial membrane proteins. Integrating both analyses the authors generate a master map of the submitochondrial protein distribution.

This is an excellent study. The experiments presented in this manuscript are rigorously done, clearly presented and multiple approaches were used to validate the results. Besides its technical quality, the study has high significance and provides unprecedented overview on submitochondrial protein localization. The fact, that databases contain a large amount of wrong information, which stems from limited experimental data, I find that this is a very important contribution to the field of molecular cell biology, which will hopefully be used to fix the database shortcomings. The study will be of broad interest and fully suitable for publication in Nature communications.

> We thank the reviewer for this positive comment on our study.

I have only a few minor points:

1. In Fig. 4e Vogtle et al. determine the peptidase activity of Prd1 claiming this protein is a presequence peptide degrading enzyme. However, in the assay they don't include negative controls (e.g. not degradable protein). Such a control would be required.

> We have now included a new experiment (new Fig. 4e) that shows that a full-length mitochondrial matrix protein (tested by addition of the radiolabelled Cox4 preprotein) is not degraded by Prd1. Moreover, we included a further control that shows that in the presence of a Prd1 variant (Prd1^{E502Q}) - with a point mutation in the catalytic site - presequence peptide degradation activity is not observed fully supporting our findings.

New Figure 4e: Degradation assay of Cox4 presequence peptide (lanes 1-12) and Cox4 precursor protein (lanes 13-25) in the absence (Ctrl., control) or presence of cell-free translated Prd1^{WT} and the variant Prd1^{E502Q}.

2. It would strengthen the manuscript, if the authors would include a carbonate extraction analysis of some of the dually localized proteins that are clustered in the ambiguous region (Suppl. Fig. 3).

> We have included new immunoblots showing carbonate extraction analysis for three different proteins that were assigned to the ambiguous region of the mitochondrial proteome. These are included in the new Supplementary Fig. 6a.

New Supplementary Fig. 6a. Immunoblot analysis of Mcr1, Ape2 and Ygr266w after carbonate extraction. P, pellet; SN, supernatant.

3. In Suppl. Fig. 2 the authors display clusters of three different classes of proteins. Few of them belong to the integral membrane cluster, e.g. Mba1 or Atp7 are well known peripheral proteins. The authors should comment on this in the text.

> In the new Supplementary Fig. 7 we compare the current model of the yeast ATP synthase structure with all components identified in this study, showing a remarkable agreement with the published models. Atp7, the subunit d of the ATP synthase, is indeed the only protein for which our data do not fit with the current model (see also comment to Reviewer 3 for Supplementary Fig. 7). However, we could not find a reference in the literature, in which the membrane topology of yeast Atp7 was systematically analyzed by carbonate extraction. We therefore now discuss the possibility that Atp7 might be integrated more deeply into the inner mitochondrial membrane than so far anticipated.

For Mba1 it has been reported by Preuss *et al.* (2001) that a major fraction (similar to the integral outer membrane protein Tom70) remains in the pellet after carbonate extraction (see Fig. 1b in the publication by Preuss *et al.*) fully supporting our findings for this protein.

4. On page 3, 17 and 20 the authors state that the SILAC ratios were determined from four biological replicates, although it seems from the Supplemental Table 1 that there are only 2 data sets. It would be good to clarify in the text that the SILAC ratio were defined from 2 biological replicates: 2 forward experiments (e.g. SNheavy/Pelletlight) and 2 reverse experiments (e.g. SNlight/Pelletheavy) for each condition, sonication and carbonate.

> We thank the reviewer for this hint. We have clarified this on pages 3, 17 and 20 and followed the suggestion of the reviewer how to describe the generation of the four data sets more clearly.

5. In Figure 3b the authors analyzed the subcellular distribution of selected proteins by Western analyses. Ape4 in Mito. and S100 fractions migrate differently. Is there an explanation for this? Lsp1 runs as a double band. Only in the P100 it appears to be solely the faster migrating form. Is this protein processed?

> The reviewer raises the interesting possibility of a processing of the two proteins Ape4 and Lsp1. Both do not seem to be proteolytically processed upon import of radiolabeled precursors into mitochondria (see Fig. 3f for Ape4 and the new Fig. R1c for Lsp1). We therefore would rather speculate about other potential post-translational modifications. Interestingly, the Lsp1 precursor shifts to a higher molecular weight upon incubation with isolated mitochondria (resulting in two bands similar to endogenous Lsp1 (Fig. 3b)). This might be due to e.g. phosphorylation, which may occur at the outer mitochondrial membrane. To exclude the possibility that one of the two Western blot bands might be an unspecific cross reacting band we generated a mitochondrial and an S100 fraction from wildtype, *ape4* Δ and *lsp1* Δ cells. For each protein both bands disappeared in the respective deletion strains (Figure R1a, b).

Figure R1: (a) and (b): Analysis of Ape4 and Lsp1 immunosignals in mitochondrial and S100 fractions from wildtype cells and respective deletion strains. (c) In organello import assay of radiolabelled Lsp1 precursor into isolated mitochondria.

6. Typos:

e.g. page 7, ref. 29 (dot); page 7, protein8 (superscript);
page 5, Suppl. Table 2 (instead of S2);
legend of the Supplementary Figure 2 (instead of S2).

> We thank the reviewer for the thorough reading and corrections. We have corrected the indicated typos in the revised manuscript.

7. On Page 3 the authors use the term “SILAC-labelled “ this appears to me as a tautology as SILAC stands for Stable Isotope Labelling in Cell Culture. This could be easily fixed.

> We have corrected this on page 3 to the less misleading term “stable isotope labelled yeast cultures”.

8. Some entries in the list of references are not formatted correctly:

e.g. page 11, reference 11: et al. (Italic);
page 13, reference 34: et al. (Italic); page 13, reference 44: 55 (Bold);
page 25, reference 54: 44 (Bold).

> We have corrected these formatting errors and would like to thank the reviewer again for this careful reading and correction.

Reviewer #2 (Remarks to the Author):

This paper concerns the mapping of proteins to sub-mitochondrial compartments, using organelle enrichment followed by submitochondrial fractionation on heavy and light SILAC labelled yeast cultures. The authors use well-established methods to sub-fractionate the mitochondria, including carbonate washing and sonication to obtain soluble protein and integral/peripheral membrane fractions. This results in a number of scores, which the authors use to estimate the enrichment of proteins in each mitochondrial sub-compartment (integral membrane, peripheral membrane or soluble) and leads to a map of the organelle in which different sub-compartment resolve from each other and form distinct clusters, with multiply localised proteins contained in the intervening space. The use of the ratios to estimate enrichment and infer sub-mitochondrial localisation of proteins is quite a nice approach to take. The authors also carry out comparison using label-free proteomics (spectral counts) to demonstrate enrichment of candidate novel mitochondrial proteins from their initial map, in the mitochondrial fraction relative to a whole-cell extract. This gives also associated ratios from which one can estimate the relative enrichment of a protein in the mitochondrial fraction and infer its localisation.

> We appreciate the reviewer's positive comment on our novel approach to infer submitochondrial protein localization.

The authors demonstrate enrichment of some of their predicted novel mitochondrially localised proteins, using crude differential centrifugation (crudely separating microsomes, mitochondria and cytosol), which shows enrichments of mitochondrial proteins as they expect and lack of enrichment of non-mitochondrial proteins. They further demonstrate import of some of their putative novel mitochondrial proteins from their SILAC label-based sub-mitochondrial map (integral membrane vs. soluble vs. peripheral membrane) using radioactive in organello import assay. They select a subset of proteins from either the peripheral, soluble or ambiguous regions of their map, allow import and perform mitochondrial sub-fractionation by a protease protection assay. This is followed by SDS-PAGE and autoradiography, to demonstrate whether radioactive proteins are been imported. A subset (12 proteins) of their predicted novel 206 mitochondrial proteins are biochemically validated using this method, (3 predicted ambiguous, 5 predicted soluble, and 4 predicted peripheral membrane proteins) (page 6 of the article). This represents less than 10% of the putative proteins and it would have been good to see additional proteins validated in this manner. Further, might it be appropriate to validate some of these observations in their system using a visual method such as super-resolution microscopy (e.g. STED) or immunogold staining (Griparic et al., 2004; Wolff et al., 2014), searching for co-localisation of the putative mitochondrially-localised proteins with some of the proteins from the authors' reference set of mitochondrially-localised proteins. Additionally the authors could consider determining protein localisation in their system using strains from the Yeast

GFP clone collection as has been performed previously (Breker et al., 2013; Chong et al., 2015; Déneraud et al., 2013; Huh et al., 2003), rather than relying on this import assay. This would offer more information regarding potential multiple localisations than can be observed from the import assay or differential centrifugation, and would even directly show visually that the protein is localised in the mitochondrion.

> Our intention in this study was to convincingly show that the novel candidate proteins are

localized in mitochondria. For this *in organello* import experiments are the gold standard in the field, particularly, because preproteins destined to inner mitochondrial compartments require the membrane potential across the inner membrane. Such a dependency is unique for mitochondrial preproteins and therefore the dependency on the membrane potential is a highly reliable biochemical evidence for mitochondrial protein localization (Steinmetz *et al.*, Nat. Genet. 2002; Sickmann *et al.*, PNAS 2003; Prokisch *et al.*, PLoS Biol 2004; Vögtle *et al.*, Cell 2009; Ieva *et al.*, Nat. Commun. 2013; Schulz *et al.*, TICB 2015; Stroud *et al.*, Nature 2016). We have outlined in the manuscript that subcellular and suborganellar localization of proteins by tagging (e.g. GFP as proposed by the reviewer) often interferes with the complicated import and sorting machineries in the two mitochondrial membranes. We have listed several examples in the manuscript as well as in our first mitochondrial proteome paper (Sickmann *et al.*, 2003) in which GFP tagging leads to cellular mislocalization. We hope this clarifies that particularly for mitochondrial proteins imaging of tagged proteins is less suited for validation than biochemical fractionation and *in organello* import. A further issue arises for multiple localized proteins such as Sod1, Num1 or Ala1. These dually distributed proteins could not be localized to mitochondria by GFP tagging and microscopy. When searching the GFP clone collection, as suggested by the reviewer, such dually localized proteins (all of which are well established mitochondrial proteins) were annotated e.g. as cytosolic and nuclear (Sod1-GFP, Ynk1-GFP or Cox17-GFP), cytosolic (Ala1-GFP) or punctate structures (Num1-GFP). No signals for a mitochondrial localization were detected (the data can be obtained from <http://yeastgfp.yeastgenome.org/>). Also many well established mitochondrial proteins such as Qcr6, Atp12, Cyc7 were localized to the cytosol as GFP fusion proteins, whereas the mitochondrial proteins Tim17-GFP and Mdm35-GFP were assigned as cytosolic and nuclear. For none of them the mitochondrial localization was identified (Huh *et al.*, 2003). We are therefore convinced, that *in organello* import, as applied in this and numerous other studies, is the method-of-choice to validate mitochondrial localization. We have additionally performed import reactions for two further candidate proteins, Smm1 and Ygr053c, that are now included as further validated proteins in Figure 3d (see new panel of Fig. 3d below).

The authors also undertake to distinguish between integral membrane proteins of the outer membrane and inner membrane using highly purified outer membrane vesicles from a light SILAC labelled culture and total membranes isolated by sodium carbonate wash from a heavy SILAC labelled culture. They work out SILAC ratios to characterise enrichment in the outer membrane fraction relative to total membranes and infer the presence and thus localisation of integral membranes from either the inner membrane or the outer membrane. This analysis must be contingent on obtaining highly pure outer membranes. The method used to obtain such outer membrane vesicles is from a paper published in 2006 by one of the authors (their reference 29). The characterisation of the outer membranes in that paper was performed by excision of spots from a 2D-PAGE gel of isolated outer membrane and characterisation of the spots using mass spectrometry. There is no validation or demonstration as to the purity or enrichment of their outer membranes in the current article. I think that this should be demonstrated by some method (e.g. utilising immunoblotting, probing with a panel of antibodies raised against integral inner or outer membrane proteins) to demonstrate such

enrichment, before any conclusions are drawn as to the localisation of any protein to either mitochondrial membrane.

> We fully agree with the reviewer that only an outer membrane fraction, which is largely devoid of inner membrane proteins is suited for the analysis undertaken in our study. We therefore show by immunoblotting of three outer membrane and three inner membrane marker proteins that the OM fraction is largely devoid of IM proteins (Fig. 4b). We furthermore show an additional quality control blot for the purity of the OM fraction in the new Supplementary Fig. 8 as requested by the reviewer. In total, we have evaluated eight integral outer membrane proteins (Tom40, Por1, Mim1, Msp1, Mcr1_{OM}, Tom70, Tom22 and OM14) and seven integral inner membrane proteins (Tim23, Sdh3, Tim21, Tim50, Tim54 and AAC/Pet9) by Western blotting. All tested IM proteins are only present in the mitochondrial fraction but absent in the OM fraction (Fig. 4b and Supplementary Fig. 8).

New Supplementary Fig. 8. Immunoblot analysis of mitochondrial proteins from various subcompartments in purified mitochondria (Mito.) and purified outer membranes (OM). The analysis included integral outer membrane proteins and integral inner membrane proteins as requested by the reviewer. Proteins from the inner mitochondrial compartments (including the IMS form of Mcr1 and the matrix localized proteins Mdh1, Aco1 and Mge1) are virtually absent in the outer membrane fraction.

In general I think that the results are impressive and show good clustering of the sub-mitochondrial proteins (in Supplementary Figures 1 and 2), but the methods used to lyse and fractionate the cells, and the authors' lack of demonstration of some of the data, are somewhat concerning. Firstly, the method used to lyse the cells is not stated, and the results of this study are highly contingent on the presence of highly enriched mitochondria which should be intact until such time as they are sub-fractionated. Further, the authors' statement that they "highly purified" their mitochondria and confirmed purity by western blotting is concerning as the western blots are not shown in this manuscript.

> We thank the reviewer for these positive comments on our results and the quality of our suborganellar proteome. We have included in the revised version new experimental data that show the high purity and intactness of our samples. For details please refer to the next paragraph. We also added a more detailed description to the Method section on how the cells were lysed and fractionated.

How do we know that the mitochondria are pure or enriched to a high enough degree? How do we know whether the mitochondria are intact and not damaged? Further their methods

section seems to suggest that they isolate only the interphase of the density gradient used for mitochondrial purification. They assume this interphase to contain the highly pure mitochondria, but nothing else. How can they be sure that there is not more than one population of mitochondria unless they sample protein from all parts of their density gradient? It is also somewhat concerning that the cells are lysed and then frozen at -80°C before thawing and performing cellular fractionation. Freeze-thaw is damaging to proteins (Cao et al., 2003) and often used to lyse cells suggesting that it can be damaging to organelles and subcellular structures. This supports further that the authors should more thoroughly analyse their whole density gradient, and demonstrate this analysis (by western blotting or other means), before they make any assumptions as to the purity of their mitochondrial preparation and draw any conclusions from it.

> We now provide in the new Supplementary Fig. 1a a detailed immunoblot analysis of cellular marker proteins for our highly purified mitochondria, showing the strong enrichment of mitochondrial proteins and the virtual absence of other cellular markers. We would also like to point out that we have developed this particular purification protocol to obtain highly pure yeast mitochondria with the actually highest purification grade that has been achieved to date (Meisinger *et al.*, 2000; 2006). The protocol is the standard in the field, as demonstrated by more than 200 citations. It was also the basis for all our previous proteomic studies including the deciphering of the entire mitochondrial proteome (Sickmann *et al.* PNAS 2003), the first mitochondrial phosphoproteome (Reinders *et al.*, MCP 2007), the first mitochondrial N-Proteome (Vögtle *et al.*, Cell 2009) and the first outer membrane phosphoproteome (Schmidt *et al.*, Cell 2011).

New Supplementary Fig. 1a. Immunoblot analysis of total yeast cells, crude and highly purified mitochondria revealing the high purity of our mitochondrial isolations. ER, endoplasmic reticulum; PM, plasma membrane; Vac., vacuole; Per., peroxisome.

We now also provide a detailed quality control analysis (new Supplementary Fig. 1b) of the mitochondrial intactness by testing the accessibility of externally added Proteinase K to inner mitochondrial compartments. Neither freshly isolated nor frozen (at -80°C) organelles became leaky and damaged as shown by the strong protection of the intermembrane space against Proteinase K treatment.

New Supplementary Fig. 1b. Control of the intactness of purified mitochondria. Samples were subjected to iso- or hypoosmotic conditions and treated with Proteinase K (20 $\mu\text{g}/\text{ml}$ for 10 min on ice followed by addition of 1 mM PMSF). Unlike osmotic rupture of the outer membrane (hypo-osmotic condition) mitochondria from iso-osmotic conditions are intact as revealed by the protection of proteins exposed to the intermembrane space (Tim50 and Tim21) against externally added Proteinase K. Mitochondrial integrity was not changed neither upon freezing (middle panel) nor gradient purification (right panel).

The gradient purified mitochondria are collected from the 32%/23% sucrose interface of the step gradient where they exclusively migrate (see also Meisinger *et al.*, 2000 and 2006, where this purification protocol was published). In Fig. R2 we show a picture of the sucrose gradient after centrifugation. The light brown mitochondria are clearly visible in the gradient. Exactly this fraction was recovered for our analysis.

Fig. R2: Sucrose gradient purification of mitochondria. Crude mitochondria were loaded on top of the three-step sucrose gradient and recovered after centrifugation from the 32%/23% sucrose interface (light brown band).

The authors construct a reference set of mitochondrial proteins, which they use to demonstrate separation achieved by their sub-fractionation approach. This reference set is plotted on the map in Supplementary Figure 1, and represents members of each of the sub-organellar niches. The authors already say, however, in their introduction that the assignment of proteins to mitochondria is problematic and a lot of the annotation to the organelle is based on data which they do not trust. How exactly was this reference set constructed? What were the exact criteria?

> The reference set was constructed based on known mitochondrial proteins for which a detailed localization including solubility and/or membrane association or membrane

integration has been experimentally shown. Each protein was manually reviewed using the original literature. A paragraph explaining this has been included in the Method section.

In Supplementary Figures 1, 2 and 3 (the mitochondrial map figures), how are the ellipses representing the clusters constructed? They seem slightly arbitrary as there do not seem to be bounds for the scores used to define their clouds. What governed the scores which were used to define membership of a specific part of the mitochondrion? How can they therefore be used to ascertain whether the proteins contained within are part of these sub-structures? How do they know that their clouds encompass the entirety of the specific sub-structure? How do they know that the clouds are elliptical and not some other shape unless they apply some sort of confidence metric or bounds to these data? I would have expected some statistical tests to be applied to determine the boundaries of the clusters and some estimate of the false discovery rates of assignment to the resulting clusters. When the authors have assigned some sort of confidence metric to their assignments, it may be useful to validate some of their specific localisation predictions. This would be performed on a subset of their high and low confidence novel proteins which are predicted to localise to each of the specific sub-mitochondrial clusters, and could include validation by other means such as super-resolution microscopy.

> As suggested by the reviewer, we applied a statistical approach to model the clusters for integral, peripheral and soluble proteins, based on the set of well-known reference proteins and their distribution. We modeled the distribution of the data points as a mixed model of three multivariate normal distributions. To compute the parameters (mean and covariance matrix) for every model, we used the ordinary maximum likelihood estimators. Here, the boundary of a two-dimensional multivariate distribution results in an elliptical shape. We chose a cumulated density threshold of 85% to visualize the boundaries. For each protein we considered the probability to be represented by any of the three generated statistical models. We had a substantial agreement between the original and the novel statistical clusters, such that ~95% of all proteins were assigned to the same group (i.e. soluble, peripheral, integral or ambiguous). The statistical likelihoods were added to supplemental table 4, the corresponding clusters are depicted in Supplementary Figs 4 and 5, and the method is described in detail in the methods section.

New supplementary Figs 4 and 5. Distribution of the reference protein set P_L (upper panel) and all proteins $P_L + P_U$ (lower panel). To indicate memberships, all 3 color channels RGB were multiplied by their corresponding probabilities for every protein. The model boundaries represent 80% of each density. Integral membrane proteins (green); peripheral membrane proteins (orange); soluble proteins (blue).

Furthermore, we analyzed and included now three additional protein complexes (ATP synthase and MICOS of the inner membrane and the protein import machineries of the outer membrane) with established topologies, all of which show a strong agreement with our clusters (new Supplementary Fig. 7) fully supporting the high quality of our landscape of submitochondrial protein distribution.

New Supplementary Fig. 7. Correlation of our data with the known submitochondrial localization of components of the ATP synthase (upper panel), the mitochondrial contact site and cristae organizing system (MICOS, middle panel) and the import machineries in the mitochondrial outer membrane (translocase of the outer membrane, TOM complex; mitochondrial import machinery, MIM complex; sorting and assembly machinery, SAM complex; lower panel) (Devenish *et al.*, 2000; Mick *et al.*, 2011; Lytovchenko *et al.*, 2014; Pfanner *et al.*, 2014)

The authors observe in the “integral membrane” fraction in each of their mitochondrial maps, the yeast plasma membrane ATPases Pma1p and Pma2p. They describe these proteins as the “most frequent contaminants in proteomics studies” citing their own papers, in which these proteins were contaminants in mitochondrial preparations. In these papers the reason given for contamination by these proteins is that they are easily accessible to tryptic digestion. In one such paper, this statement is backed up with an observation from the work of Washburn *et al.* (Washburn *et al.*, 2001), which itself does not in fact say that these proteins are contaminants in proteomics studies. I therefore do not think that this is a particularly strong argument. The authors say that the supposed accessibility of Pma1p and Pma2p to tryptic digest enables these proteins to be detected even in “tiny amounts”. The authors thus discount them from further analysis. If this were the case, would it not be expected to be true for other proteins which exhibit equal accessibility to tryptic digest? The presence of these proteins in their preparations could be indicative of the purity or degree of enrichment of their mitochondrial samples and cast doubt on some of their other

observations regarding “novel” mitochondrial protein assignments. To make such a statement, it is argued that they should find evidence for this observation from other sources.

> We apologize for this misunderstanding. Indeed, the sentence should read “The most frequent contaminant in mitochondrial proteomic studies” and has been corrected. This is supported by a variety of yeast mitochondrial proteome studies from our and other labs, which are listed in the references.

In summary, the manuscript describes a huge amount of work to determine the submitochondrial proteome and the data presented have the potential to be of high utility to researchers. As currently presented the study falls short in two main areas; demonstration of the purity of mitochondria achieved and their integrity after purification/enrichment, robust statistical analysis of the resulting data. Without these shortcomings being addressed, the conclusions made in this study cannot be fully supported and hence the manuscript is not yet ready for publication in Nature Communications.

> We hope we could clarify all points raised by the reviewer, in particular by the new experiments demonstrating the purity and integrity of our samples and by including the statistical analysis as pointed out above.

Specific smaller comments:

The order of the introduction seems a little odd with the results summarised before being put into context, i.e. previous proteome maps of the mitochondrial being far from complete or well resolved into subcompartments.

> We have now clarified in the introduction part that the previous in-depth proteomic maps of mitochondria were performed on total purified organellar fractions.

In the introduction (page 1), the authors say that 986 proteins were assigned, but do not say whether they were assigned to a specific sub-organellar location or to the organelle in general. This should be made clearer.

> We apologize for this unclear description. We have clarified now what exactly this number means and differentiate between the number of proteins assigned according to their biophysical properties and the number of proteins, which could be finally assigned to their exact submitochondrial compartment.

On page 3 (results) they say that they isolated highly pure light and heavy mitochondria, but it is unclear from this text whether they mean light membranes and heavy membranes or light SILAC labelled and heavy SILAC labelled mitochondria. This should be made clearer.

> We thank the reviewer for this note. On page 3 we describe the analysis of entire mitochondria subjected to carbonate or sonication treatment. This is different from the analysis of the membrane fractions from outer membrane and total mitochondria, applied to differentiate outer from inner membrane proteins (Fig. 4). We have carefully checked that the description of the respective material used for the analysis in each subset of the manuscript is correctly explained now.

On page 4, they say that a subset of proteins in their map are localised to the “correct” clusters, but they do not know that their clusters are correct. “Expected” would be better to use here, as in the absence of further validation they do not know whether their assumptions are correct.

> We thank the reviewer for this suggestion and changed “correct” now to “expected”.

They also make a sweeping statement that the mitochondrial proteome consists of the proteins that they have detected in their study. This is a bit of a generalisation, as they could say that “their” mitochondrial proteome consists of these proteins. They do not know whether their proteome is exhaustive, what the level of contamination is or whether they are not sampling a fraction of the mitochondrial proteome.

> We thank the reviewer for this comment and agree that it might be more appropriate now to state “we identified 321 integral membrane proteins, 258 peripheral membrane proteins, 226 soluble proteins...” instead of “the proteome consists of...”. We have changed this in the revised manuscript.

On page 5, they mention that the “majority” of known mitochondrial proteins show low yeast/mito ratios using their spectral counting approach, but it is not clear what this number represents as the supplementary table referenced is not available. I think that this should be changed to a definite value, even if it is available in a supplementary table.

On page 22, under Quantitative Comparison of total yeast vs. total mitochondrial proteomes, the authors state what their chromatography solvent A is, but not solvent B. This should be stated.

> We apologize for this inconvenience, which might have been due to a pdf conversion error during uploading of the original excel tables. We have now uploaded the supplementary tables as fully accessible excel files and refer to Figure 3b where we propose the definition of three different localization categories (Yeast/mito ratio <1, between 1 and 10 and >10). We further thank the reviewer for the thorough reading and we have added the composition of the solvent B now in the revised manuscript.

On page 24, the authors state that they digitally altered their western blots to remove non-relevant bands. It is argued that this is neither appropriate nor acceptable and that the scanned blots should be included in their entirety.

> We are sorry that this phrasing was misleading. We entirely follow the strict standards of the community (as e.g. published by the Nature Journals (<http://www.nature.com/authors/policies/image.html>) and Journal of Cell Biology, Rossner & Yamada, JCB 166,11 (2004)). It is absolutely common standard in the field that only the regions of interest (of sometimes large films/blots and autoradiographs) are shown. We also follow the standards of Nature Communications by providing the entire set of original data (see all original immunoblots and autoradiography scans which have been included now in the supplement file). To avoid any misunderstandings the description in the Methods part now reads “To show regions of interest blots and autoradiography scans were digitally processed”.

In their Online Methods section, the concentration of yeast nitrogen base in their minimal media used for SILAC seems to be 10 times too high at 6.7% (w/v) (normal concentration for

minimal media is 0.67% (w/v)) (Sherman, 2002). Further, the OD₆₀₀ to which they grow their cells has quite a wide range (0.7-1.5), which, can encompass different growth phases of yeast (from early to mid-exponential phase). It is unclear why they use such a wide range of optical densities and this should be explained, as it might be expected that the proteins could change localisation based on growth phase.

> We thank the reviewer for detecting this mistake. We have now changed the concentration of the yeast nitrogen base in the Method section to 0.67%. We now specify the exact OD₆₀₀ of the yeast cultures used in this study that were obtained between an OD₆₀₀ of 0.7 – 1.0 for the generation of highly pure mitochondria and outer membrane vesicles. Mitochondria for in organello experiments were isolated from yeast cultures with an OD range between 0.7 and 1.5. The OD did not influence targeting of radiolabeled precursors into mitochondria.

References

- Breker, M., Gymrek, M., and Schuldiner, M. (2013). A novel single-cell screening platform reveals proteome plasticity during yeast stress responses. *J. Cell Biol.* 200, 839–850.
- Cao, E., Chen, Y., Cui, Z., and Foster, P.R. (2003). Effect of freezing and thawing rates on denaturation of proteins in aqueous solutions. *Biotechnol. Bioeng.* 82, 684–690.
- Chong, Y.T., Koh, J.L.Y., Friesen, H., Duffy, K., Cox, M.J., Moses, A., Moffat, J., Boone, C., and Andrews, B.J. (2015). Yeast proteome dynamics from single cell imaging and automated analysis. *Cell* 161, 1413–1424.
- Dénervaud, N., Becker, J., Delgado-Gonzalo, R., Damay, P., Rajkumar, A.S., Unser, M., Shore, D., Naef, F., and Maerkl, S.J. (2013). A chemostat array enables the spatio-temporal analysis of the yeast proteome. *TL - 110. Proc. Natl. Acad. Sci. U. S. A.* 110 VN-, 15842–15847.
- Griparic, L., van der Wel, N.N., Orozco, I.J., Peters, P.J., and van der Bliek, A.M. (2004). Loss of the Intermembrane Space Protein Mgm1/OPA1 Induces Swelling and Localized Constrictions along the Lengths of Mitochondria. *J. Biol. Chem.* 279, 18792–18798.
- Huh, W.-K., Falvo, J. V., Gerke, L.C., Carroll, A.S., Howson, R.W., Weissman, J.S., and O’Shea, E.K. (2003). Global analysis of protein localization in budding yeast. *Nature* 425, 686–691.
- Sherman, F. (2002). Getting started with yeast. *Methods Enzymol.* 350, 3–41.
- Washburn, M.P., Wolters, D., and Yates, J.R. (2001). Large-scale analysis of the yeast proteome by multidimensional protein identification technology. *Nat. Biotechnol.* 19, 242–247.
- Wolff, N.A., Ghio, A.J., Garrick, L.M., Garrick, M.D., Zhao, L., Fenton, R.A., and Thévenod, F. (2014). Evidence for mitochondrial localization of divalent metal transporter 1 (DMT1). *FASEB J.* 28, 2134–2145.

Reviewer #3 (Remarks to the Author):

This manuscript, submitted as a Resource, attempts to generate a map of yeast submitochondrial protein localization via a series of biochemical treatments and fractionations coupled with quantitative mass spectrometry techniques. By and large, the experimental techniques are all carefully and rigorously performed and the resulting dataset is high-quality, well organized, and informative. The current version risks overreaching in certain areas and could provide additional guidance/rationale on certain parameters. However, given that the points below are addressed, this work will serve as a very useful resource for the mitochondrial community and should make for a timely Nature Communications publication.

> We thank the reviewer for these very positive comments and we have addressed the indicated points as outlined below.

Major points

1. In reading the introduction, one would think that considerable efforts to map sub-mitochondrial proteomes had not already been done. In particular, I find that Alice Ting's APEX2 work to be conspicuously absent. This work is later referenced in the Discussion, but it should be introduced earlier in the paper. Additionally, an effort should be made to compare the results of these efforts for proteins conserved between yeast and mammalian systems. Such analyses would be useful whether or not the data proved to be consistent.

> The APEX technology developed in the Ting lab is a highly elegant procedure to profile proteins, which are localized in close proximity of the targeted bait protein. This implies that the method rather deciphers the toponome of a dedicated organellar compartment. Consistent with this, the approach cannot differentiate (e.g. using a matrix targeted bait) whether identified hits are soluble matrix proteins, peripherally attached inner membrane proteins or integral inner membrane proteins with domains facing the matrix. Our approach, however, is exactly tackling this question. It is therefore quite difficult to compare data from APEX approaches with our datasets. We would here compare apple and oranges and we are afraid that this might not be of much help to the reader and we have tried to point this out in the Discussion part.

The dilemma of comparing our and the APEX data is illustrated in the cartoon below (Figure R3) depicting a figure of the original paper by the Ting lab (Rhee et al., Science 2013; Fig. 2C) that shows proteins of the human respiratory chain complexes of the inner membrane which were identified as matrix proteins. All proteins in red were found to interact with the matrix targeted bait protein and were therefore annotated as 'matrix'. In our approach, however, these proteins are classified as peripheral and integral inner membrane and soluble proteins, as shown below. Due to reasons outlined above that the APEX data describe the toponome/proximity of a mitochondrial protein rather than its actual physical presence in a distinct subcompartment a comparison of these data sets would imply a low consistency of the Ting data. As requested by the reviewer, we now introduce the work of the Ting lab earlier in the introduction part of our manuscript.

Figure R3: Upper panel: Identified matrix proteins of respiratory chain complexes in human mitochondria via APEX technology from Rhee *et al.* (2013, Fig 2C of this publication). All proteins in red were annotated as matrix proteins due to their interaction with a matrix targeted bait protein. Lower panel: Classification of the

components of the yeast respiratory chain complexes into the distinct mitochondrial subcompartments via our landscape of submitochondrial protein distribution.

2. The use of the yeast/mito ratio for defining novel mitochondrial proteins is underdeveloped and oversimplified as currently depicted. Of course, any co-purifying organelle fraction would also score well by this analysis. What kind of yeast/mito scores to well-established proteins from other organelles receive, and how do these compare to the novel proteins? Also, while the validation of the selected novel proteins is convincing, it is not clear why they were chosen.

> We thank the reviewer for this comment. It is indeed absolutely essential for proteomic studies that the isolated organelles are highly pure and largely devoid of other contaminating cell compartments. As shown above in the response to reviewer 2 (see also new Supplementary Fig. 1a, page 7 of this letter) we now provide a detailed analysis of the purity of the mitochondria used in this study. Moreover, we have searched for the classical set of well-established cellular marker proteins, that had been used e.g. by Huh et al. (Nature 425, 2003) for the generation of the YeastGFP localization database, as markers for co-localization and in order to provide yeast/mito scores as requested by the reviewer. Of the 12 subcellular markers 10 were not identified in our isolated mitochondria but in total yeast samples and would therefore have a theoretically infinite Yeast/Mito ratio; they can therefore be considered as basically absent in our samples (see Table R1 below). For the endosomal marker Snf7 a Yeast/Mito ratio of 6.6 was found indicating either an eclipsed localization or contamination consistent with the proposed classes in our manuscript and illustrated by several examples (e.g. Sod1, Num1, Ala1). Erg6, a marker for lipid particles, was found in the outer mitochondrial membrane of our data set. Notably, Erg6 was shown to localize not only to lipid particles but also to the ER and mitochondria (McCammon *et al.*, 1984) and constitutes a physical interaction between mitochondria and lipid droplets (Pu *et al.*, 2011). Consistently, this protein scores in our analysis with a Yeast/Mito ratio of > 1 indicating a further cellular localization.

Localization	Accession	Gene	yeast/mito
Actin	YDR129C	SAC6	n.d.
Early Golgi/Cop1	YDL145C	COP1	n.d.
Endosome	YLR025W	SNF7	6.6
ER to Golgi vesicle	YLR208W	SEC13	n.d.
Golgi apparatus	YEL036C	ANP1	n.d.
Late Golgi/clathrin	YGL206C	CHC1	n.d.
Lipid particle	YML008C	ERG6	1.4
Nucleus	YPL170W	DAP1	n.d.
Nucleolus	YLR197W	SIK1	n.d.
Nuclear periphery	YFR002W	NIC96	n.d.
Peroxisome	YDR329C	PEX3	n.d.
Spindle pole	YKL042W	SPC42	n.d.

Table R1. Analysis of cellular marker proteins used as standards for the yeast GFP localization database (Huh *et al.*, Nature 2003). n.d., not detected in our highly purified mitochondria.

Furthermore, we searched the literature for the most typical organellar marker proteins in yeast and based on our quantitative yeast proteome data we selected the two most abundantly expressed markers per organelle (only one found for endosomes) (Rieder & Emr, 2001). We searched for the respective yeast/mito ratios to assess potential contamination by these organelles (see Table R2 below). Notably, 11 out of 15 marker proteins could not be

detected in our purified mitochondria, despite measuring a total of 20 fractions by nano LC-MS on a Q-Exactive Plus mass spectrometer. The other four proteins comprised two ER markers with high yeast/mito ratios of 8.2 and 7.8, as well as two peroxisomal markers with ratios above 100, as summarized in the table below (n.d. = not detected in mitochondria). Summarizing all these data we conclude that the extent of contamination in our dataset is remarkably low and that other cell organelles do not appear to co-purify with mitochondria which would result in a yeast/mito score < 1.

Organelle Marker	Accession	Gene	yeast/mito
Cytoplasm	YNL241C	ZWF1	n.d.
Cytoplasm	YDL022W	GPD1	n.d.
endosome	YOR036W	PEP12	n.d.
ER	YCL043C	PDI1	8.2
ER	YJL034W	KAR2	7.8
golgi	YEL042W	GDA1	n.d.
golgi	YGL038C	OCH1	n.d.
nucleus	YBR010W	HHT1	n.d.
nucleus	YDL014W	NOP1	n.d.
peroxisome	YDR256C	CTA1	>>> 100
peroxisome	YIL160C	POT1	413.9
vacuoles	YBR127C	VMA2	n.d.
vacuoles	YDL185W	VMA1	n.d.
vesicles	YGR167W	CLC1	n.d.
Vesicles	YGL206C	CHC1	n.d.

Table R2. Comparison of typical organellar markers (Rieder & Emr, 2001). The two most abundant proteins (as measured in the total yeast sample) per organelle were selected. n.d., not detected in our highly purified mitochondria.

For testing of the mitochondrial localization of our candidate proteins via in organello import we had to establish in vitro translation conditions for each candidate to get a sufficient amount of radiolabelled precursors. The proteins used for import studies were the ones for which we could obtain a sufficiently hot lysate (this is usually the bottleneck for this kind of analysis). For a few (shown in Fig. 3b) we could also successfully generate specific antibodies, which were used additionally for validation. We explain this now in more detail in the results section.

Minor points

1. A short explanation should be given for a broader audience as to why some of the validated novel proteins have observable size shifts upon import and others do not

> We thank the reviewer for this suggestion and explain the size shifts of precursor proteins upon import in the figure legend to enable a direct understanding of the experiments.

2. In the CoQ reference set and corresponding Figure 2C, it should be noted that Coq8 has a single-pass transmembrane domain. Interestingly, this seems consistent with its lower SN_{son}/PEL_{son} score compared to the other Coq-related proteins.

> Thanks for the note on this interesting observation. Indeed it appears that beside the clear separation of the integral membrane protein Coq2 the remaining CoQ proteins cluster in two different regions: one (incl. Coq3, 4, 5, 6, 9 and Cat5) relatively close to the soluble domain indicating a rather loose membrane interaction and Coq8 and Coq1 rather close to the integral cloud which might indicate a more tight membrane association (Fig. 2c). We are not aware of an experimental demonstration of a TM domain in Coq8, but included this note as suggested by the reviewer.

References

- Devenish, R.J., Prescott, M., Roucou, X. & Nagley, P. Insights into ATP synthase assembly and function through the molecular genetic manipulation of subunits of the yeast mitochondrial enzyme complex. *Biochim. Biophys. Acta* **1458**, 428-442 (2000).
- McCammon, M.T., Hartmann, M.A., Bottema, C.D. & Parks, L.W. Sterol methylation in *Saccharomyces cerevisiae*. *J. Bacteriol.* **157**, 475-483 (1984).
- Huh, W.-K. *et al.* Global analysis of protein localization in budding yeast. *Nature* **425**, 686-691 (2003).
- Ieva, R. *et al.* Mitochondrial inner membrane protease promotes assembly of presequence translocase by removing a carboxy-terminal targeting sequence. *Nat. Commun.* **4**, 2853 (2013).
- Lytovchenko, O. *et al.* The INA complex facilitates assembly of the peripheral stalk of the mitochondrial F1F0-ATP synthase. *EMBO J.* **33**, 1624-1638 (2014).
- Meisinger, C., Pfanner, N. & Truscott, K.N. Isolation of yeast mitochondria. *Methods Mol. Biol.* **313**, 33-40 (2006).
- Meisinger, C., Sommer, T. & Pfanner, N. Purification of *Saccharomyces cerevisiae* mitochondria devoid of microsomal and cytosolic contaminations. *Anal. Biochem.* **287**, 339-342 (2000).
- Mick, D.U., Fox, T.D. & Rehling, P. Inventory control: cytochrome c oxidase assembly regulates mitochondrial translation. *Nat. Rev. Mol. Cell Biol.* **12**, 14-20 (2011).
- Pfanner, N. *et al.* Uniform nomenclature for the mitochondrial contact site and cristae organizing system. *J. Cell Biol.* **204**, 1083-1086 (2014).
- Preuss, M. *et al.* Mba1, a novel component of the mitochondrial protein export machinery of the yeast *Saccharomyces cerevisiae*. *J. Cell Biol.* **153**, 1085-1096 (2001).
- Prokisch, H. *et al.* Integrative analysis of the mitochondrial proteome in yeast. *PLoS Biol.* **2**, e160 (2004).
- Pu, J. *et al.* Interactomic study on interaction between lipid droplets and mitochondria. *Protein Cell* **2**, 487-496 (2011).
- Reinders, J. *et al.* Profiling phosphoproteins of yeast mitochondria reveals a role of phosphorylation in assembly of the ATP synthase. *Mol. Cell. Proteomics* **6**, 1896-1906 (2007).
- Rhee, H.W. *et al.* Proteomic mapping of mitochondria in living cells via spatially restricted enzymatic tagging. *Science* **339**, 1328-1331 (2013).
- Rieder, S.E. & Emr, S.D. Overview of subcellular fractionation procedures for the yeast *Saccharomyces cerevisiae*. *Curr. Protoc. Cell Biol.* **3**, 3.7 (2001).
- Schmidt, O. *et al.* Regulation of mitochondrial protein import by cytosolic kinases. *Cell* **144**, 227-239 (2011).
- Schulz, C., Schendzielorz, A. & Rehling, P. Unlocking the presequence import pathway. *Trends Cell Biol.* **25**, 265-275 (2015).
- Sickmann, A. *et al.* The proteome of *Saccharomyces cerevisiae* mitochondria. *Proc. Natl. Acad. Sci. USA* **100**, 13207-13212 (2003).
- Smith, P.M., Fox, J.L. & Winge, D.R. Biogenesis of the cytochrome bc1 complex and role of assembly factors. *Biochim. Biophys. Acta* **1817**, 872-882 (2012).
- Steinmetz, L.M. *et al.* Systematic screen for human disease genes in yeast. *Nat. Genet.* **31**, 400-404 (2002).
- Stroud, D.A. *et al.* Accessory subunits are integral for assembly and function of human mitochondrial complex I. *Nature* **538**, 123-126 (2016).
- Vögtle, F.N. *et al.* Global analysis of the mitochondrial N-Proteome identifies a processing peptidase critical for protein stability. *Cell* **139**, 428-439 (2009).

Reviewers' Comments:

Reviewer #1 (Remarks to the Author):

In the revised version, the authors provide a substantial amount of new data and textual adaptation to adequately address the points raised by the reviewers. In my view the revised manuscript is substantially strengthened and the concerns of the reviewers have been fully addressed. I especially appreciate that the authors clearly discuss in their letter the inappropriateness of GFP-fusions for proper localization. The misconception that such an approach provides reliable data in all cases is a severe problem for molecular cell biology. I very much support publication of this manuscript in its current version in Nature communication.

Reviewer #2 (Remarks to the Author):

The comment regarding use of fluorescent/super-resolution imaging of mitochondrial proteins was really a query as to the purity and intactness of the preparation. If the integrity and high enrichment of mitochondria in the preparation could not be demonstrated within this article, then the application of a complimentary microscopy-based method which does not rely on a high degree of enrichment of mitochondria to confirm organelle location is a sensible approach to take. Further, the suggestion of using the GFP Clone Collection, did not simply refer to comparing data with, for example, the published work of Huh et al. (2003), but rather to obtaining a subset of strains from the strain collection and performing the authors' own de novo analysis specifically for this paper, using some of the putatively mitochondrial proteins identified under the subheading "Identification of novel mitochondrial proteins". In some cases, the authors of the references listed in the original comments obtained different localisation results from each other for the same protein. In the current study of Vögtle et al., localisation results which are different from Huh et al. (2003) may therefore be expected to be obtained. A microscopy-based method (whether fluorescent tag-based, super-resolution based or based on indirect immunofluorescence) may add a further layer of information, in addition to the gold standard method described by the authors as one would be able to see not only whether the protein is localised to the mitochondrion but also whether it has another localisation and where that localisation is.

The authors' explanations regarding their choice to use the import assay for this purpose is acceptable in the revised version of the manuscript. As regards the additional import assays performed by the authors, for Smm1 and Ygr053c, the addition of these two further assays is an improvement, but this amounts to only 14 out of 206 proteins. This amounts still to less than 10% of all putative mitochondrially-localised proteins mentioned in that section of the paper. It

is further noted that the new proteins validated by this assay are only from the “soluble” protein class, and there are new validations for neither the “ambiguous” nor “peripheral membrane” classes (Figs. 3e and 3f). The authors should therefore conduct additional validity import assays for proteins from the other two classes (“ambiguous” and “peripheral membrane”). The thorough new western blot analysis of OM and IM proteins performed by the authors to demonstrate a high degree of enrichment of OM proteins in the OM fraction and depletion in the other fractions (New Supplementary Fig. 8) is now acceptable. The additional demonstration of enrichment of mitochondria relative to other organelles shown in New Supplementary Fig. 1a is also an excellent addition to the manuscript.

The attempted demonstration of intactness of purified mitochondria, whether fresh or frozen in New Supplementary Fig. 1b is a little concerning, however. It seems that the panel of antibodies which have been used in New Supplementary Fig. 1b only represents proteins which are integral to the mitochondrial inner membrane or the outer membrane (see the descriptions of the localisations of these proteins in New Supplementary Fig. 8 of the rebuttal). Thus the western blots in New Supplementary Fig. 1b seem to demonstrate only that the indicated integral membrane proteins are still within the inner and outer membranes and not that the mitochondria are still intact post-freezing. The western blots in New Supplementary Fig. 1b do not give any indication as to whether ice crystal formation on freezing has ruptured any membrane, allowing any leakiness of matrix or intermembrane space proteins. In order to demonstrate intactness the authors should perform additional experiments, but utilising antibodies directed against proteins which are known to reside in the intermembrane space and not to be integral to any membrane. A suggestion would be to use some antibodies raised against candidate intermembrane space proteins from the authors’ previous work (Vögtle et al. (2012). Intermembrane Space Proteome of Yeast Mitochondria. *Mol. Cell. Proteomics* 11, 1840-1852).

Further to this, it is unclear what “highly purified” refers to in the third panel of New Supplementary Fig. 1b. Is this referring to highly purified fresh mitochondria or mitochondria which have been highly purified after freezing? In either case, is this panel not redundant, as the information is already contained within one of the first two panels?

The comment regarding Pma1 and Pma2 being common contaminants and thus not considered in the analysis contained in this article, although discussed by the authors in the rebuttal with the additional adjective “mitochondrial”, has not been sufficiently addressed. In the rebuttal, there is a mention of this point being “supported by a variety of yeast mitochondrial proteome studies... listed in the references”. However in the revised manuscript, the studies cited as evidence for this are still the work of the authors (their references 6 and 31), which seem originally to cite the work of Washburn et al. (2001). The work of Washburn et al. makes no mention of these proteins being the most common contaminants in mitochondrial proteome studies. The original comment about this point not being very strong still stands. If this is true for these contaminant

proteins “even if they are present in tiny amounts”, due to their accessibility to tryptic digest, would it not also be expected to be true for other proteins in mitochondrial proteome studies, which exhibit equal accessibility to tryptic digest and are also present in tiny amounts? Additional evidence for this from alternative sources (a study from another group) should be proffered or alternatively an acknowledgement that the presence of appreciable plasma membrane ATPase contamination and, potentially other plasma membrane contamination (see following paragraphs), is a shortcoming of the approach taken. Moreover, additional organelle protein contamination in the authors’ preparations is still not discussed by the authors within the manuscript.

The method used in the paper for classification of proteins to specific clusters in New Supplementary Figures 4 and 5 risks overfitting of the data as the entirety of training data is used to give a static view of the mitochondrial map. There is no assessment of the performance of the method used. The authors may consider leaving some data out of the training data from New Supplementary Figure 4 and seeing how this affects the classification obtained in New Supplementary Figure 5 and whether the proteins are assigned to the same clusters, or whether their model can assign the left out training data to the expected sub-organellar cluster.

It is clear, from Supplementary Figure S2 of the originally submitted manuscript and the observations made in the paper (such as in New Supplementary Figure 7), that this method is successful in assigning a lot of proteins which are annotated in the literature as mitochondrial. However, the two discounted contaminants (Pma1 and Pma2, the plasma membrane ATPases), are “assigned” by this method to being part of the integral membrane cluster. These are clearly not integral to any mitochondrial membrane and yet are still assigned as such by this method. A thorough analysis of false discoveries in the results of this study needs to be undertaken so that it is clear what proportion of these observations are actually applicable to mitochondria. The authors must comment on false discoveries such as those already acknowledged (Pma1 and Pma2), and other contaminants in specified clusters (annotation mentioned is manually curated Cellular Component annotation from the *Saccharomyces Genome Database*), such as those listed below:

In the “soluble cluster”

Nucleoporin NUP116 (nuclear annotation).

AMS1 (vacuolar annotation).

In the “integral cluster”

IST2 and TCB3, both annotated as cortical ER and involved in cortical ER/plasma membrane tethering.

HXT2, a high affinity hexose transporter of the plasma membrane.

GAS5 and EXG2, which are both annotated as being involved in plasma membrane and cell wall-related pathways and are annotated as being localised to these locations.

YCK2, which is annotated as localising to the bud tip, mating projection and plasma membrane.

JEN1, which is annotated as localising to the plasma membrane.

At the moment, this model seems to assign both mitochondrial and contaminant integral membrane proteins to the “integral” cluster, and contaminant proteins to the soluble cluster. It is conceivable that proteins such as those mentioned above may represent novel mitochondrial proteins, but they may also represent mis-assignments and false discoveries brought about as a direct consequence of the method used in the paper to prepare the mitochondria. The authors should therefore discuss the shortcomings of the method they used for classification of proteins to each of the clusters and how they dealt with any false discoveries. Although proteins from complexes such as those displayed in New Supplementary Figure 7 are found in the expected clusters, demonstrating that the method used works to an extent, protein assignments which conflict with the literature, such as those mentioned above should not be ignored. The authors should be encouraged to perform validation of some of their specific localisation predictions such as those potential novel localisations/false discoveries mentioned above. This may be performed using other means such as the import assay described by the authors or indirect immunofluorescence, due to the authors’ aforementioned interference of a fluorescent tag with mitochondrial protein import.

Reviewer #3 (Remarks to the Author):

I have not further concerns, and I congratulate the authors on an excellent manuscript that will be of great utility to the field.

Reviewer 1

In the revised version, the authors provide a substantial amount of new data and textual adaptation to adequately address the points raised by the reviewers. In my view the revised manuscript is substantially strengthened and the concerns of the reviewers have been fully addressed. I especially appreciate that the authors clearly discuss in their letter the inappropriateness of GFP-fusions for proper localization. The misconception that such an approach provides reliable data in all cases is a severe problem for molecular cell biology.

I very much support publication of this manuscript in its current version in Nature communication.

> We thank Reviewer 1 for this very positive comment.

Reviewer 2

The comment regarding use of fluorescent/super-resolution imaging of mitochondrial proteins was really a query as to the purity and intactness of the preparation. If the integrity and high enrichment of mitochondria in the preparation could not be demonstrated within this article, then the application of a complimentary microscopy-based method which does not rely on a high degree of enrichment of mitochondria to confirm organelle location is a sensible approach to take. Further, the suggestion of using the GFP Clone Collection, did not simply refer to comparing data with, for example, the published work of Huh et al. (2003), but rather to obtaining a subset of strains from the strain collection and performing the authors' own de novo analysis specifically for this paper, using some of the putatively mitochondrial proteins identified under the subheading "Identification of novel mitochondrial proteins". In some cases, the authors of the references listed in the original comments obtained different localisation results from each other for the same protein. In the current study of Vögtle et al., localisation results which are different from Huh et al. (2003) may therefore be expected to be obtained. A microscopy-based method (whether fluorescent tag-based, super-resolution based or based on indirect immunofluorescence) may add a further layer of information, in addition to the gold standard method described by the authors as one would be able to see not only whether the protein is localised to the mitochondrion but also whether it has another localisation and where that localisation is.

> Purity and intactness of our mitochondrial preparations have now been addressed in detail and demonstrated in the revised version (see also Reviewer's comment in the next paragraph). The use of microscopy has its limitations in respect to dual or multiple localized proteins (as outlined in detail in the manuscript and the rebuttal letter). In our opinion, a re-analysis of the data

published by Huh et al. would not have yielded a different outcome and thereby further information. Moreover, it is far beyond the scope of our study to analyze the extra-mitochondrial localization of proteins. Instead our aim is to provide the community here with the first available global landscape of submitochondrial protein distribution. This is the main impact of our paper. The identification of novel candidates of mitochondrial proteins (due to the enhanced sensitivity by subfractionating the organelle) is an additional aspect of the paper and the provided Yeast/Mito ratios serve as important tool for further subcellular investigations of these proteins. We discuss in the manuscript that high Yeast/Mito ratios may indicate a multiple localized mitochondrial protein OR a contaminant (page 5 of the manuscript).

The authors' explanations regarding their choice to use the import assay for this purpose is acceptable in the revised version of the manuscript. As regards the additional import assays performed by the authors, for Smm1 and Ygr053c, the addition of these two further assays is an improvement, but this amounts to only 14 out of 206 proteins. This amounts still to less than 10% of all putative mitochondrially-localised proteins mentioned in that section of the paper. It is further noted that the new proteins validated by this assay are only from the "soluble" protein class, and there are new validations for neither the "ambiguous" nor "peripheral membrane" classes (Figs. 3e and 3f). The authors should therefore conduct additional validity import assays for proteins from the other two classes ("ambiguous" and "peripheral membrane").

The thorough new western blot analysis of OM and IM proteins performed by the authors to demonstrate a high degree of enrichment of OM proteins in the OM fraction and depletion in the other fractions (New Supplementary Fig. 8) is now acceptable. The additional demonstration of enrichment of mitochondria relative to other organelles shown in New Supplementary Fig. 1a is also an excellent addition to the manuscript.

> We thank the reviewer for these positive comments on our mitochondrial purification strategy and enrichment of the outer membrane fraction. Regarding the validation of novel mitochondrial proteins we would like to point out that this approach is depending on the generation of radiolabelled precursor proteins by an in vitro transcription and translation reaction. This limits the amount of precursors obtainable for testing in in organello import experiments. Furthermore, we are of the opinion that reaching the reviewers mark of importing 10% of all novel mitochondrial candidate proteins would not considerably strengthen our manuscript and be beyond the scope of its main aim which is the comprehensive landscape of submitochondrial protein distribution.

The attempted demonstration of intactness of purified mitochondria, whether fresh or frozen in New Supplementary Fig. 1b is a little concerning, however. It seems that the panel of antibodies which have been used in New Supplementary Fig. 1b only represents proteins which are integral to the mitochondrial inner membrane or the outer membrane (see the descriptions of the localisations of

these proteins in New Supplementary Fig. 8 of the rebuttal). Thus the western blots in New Supplementary Fig. 1b seem to demonstrate only that the indicated integral membrane proteins are still within the inner and outer membranes and not that the mitochondria are still intact post-freezing. The western blots in New Supplementary Fig. 1b do not give any indication as to whether ice crystal formation on freezing has ruptured any membrane, allowing any leakiness of matrix or intermembrane space proteins. In order to demonstrate intactness the authors should perform additional experiments, but utilising antibodies directed against proteins which are known to reside in the intermembrane space and not to be integral to any membrane. A suggestion would be to use some antibodies raised against candidate intermembrane space proteins from the authors' previous work (Vögtle et al. (2012). Intermembrane Space Proteome of Yeast Mitochondria. Mol. Cell. Proteomics 11, 1840-1852)..

> The approach taken to demonstrate mitochondrial integrity (intact mitochondrial outer and inner membrane) was the following: We incubated isolated mitochondria in an either isotonic (= Mito.) or hypotonic buffer (=Mitopl.). The hypotonic buffer leads to swelling of the mitochondria and rupture of the outer membrane thereby mimicking non-intact mitochondria. The addition of Proteinase K is then revealing if proteins normally protected by the outer membrane can be degraded. In the "Mito." sample addition of the protease does not result in degradation of proteins exposed to the IMS (Tim50 ,Tim21) or proteins attached to the IM from the mitochondrial matrix site (Cox6, which is not an integral membrane protein, but exists in a peripheral and soluble pool in the matrix). Only upon experimentally induced rupture of the OM ("Mitopl.") the IMS exposed proteins are digested. Therefore, our assay does not demonstrate the integration of proteins in the mitochondrial membranes as assumed by the reviewer, but indeed shows the intactness of our mitochondrial preparations. This approach (particularly analyzing protease accessibility of the IMS exposed domains of Tim50 and Tim21) is the standard in the field to clearly show intactness/integrity of purified mitochondria. We included now a more detailed explanation in the figure legend (Suppl. Figure 1) and cite respective publications demonstrating the validity of this approach (Chancinska *et al.*, Cell 2005 and Song *et al.*, EMBO Rep. 2014).

Further to this, it is unclear what "highly purified" refers to in the third panel of New Supplementary Fig. 1b. Is this referring to highly purified fresh mitochondria or mitochondria which have been highly purified after freezing? In either case, is this panel not redundant, as the information is already contained within one of the first two panels?

> "Highly purified" is referring to mitochondria additionally purified over a sucrose gradient after initial isolation. The treatment of these mitochondria is therefore different from the mitochondria in panel 1 and 2 (for details please see the Method section). We explain this now more detailed in the figure legend (Suppl. Figure 1).

The comment regarding Pma1 and Pma2 being common contaminants and thus not considered in the analysis contained in this article, although discussed by the authors in the rebuttal with the additional adjective “mitochondrial”, has not been sufficiently addressed. In the rebuttal, there is a mention of this point being “supported by a variety of yeast mitochondrial proteome studies... listed in the references”. However in the revised manuscript, the studies cited as evidence for this are still the work of the authors (their references 6 and 31), which seem originally to cite the work of Washburn et al. (2001). The work of Washburn et al. makes no mention of these proteins being the most common contaminants in mitochondrial proteome studies. The original comment about this point not being very strong still stands. If this is true for these contaminant proteins “even if they are present in tiny amounts”, due to their accessibility to tryptic digest, would it not also be expected to be true for other proteins in mitochondrial proteome studies, which exhibit equal accessibility to tryptic digest and are also present in tiny amounts? Additional evidence for this from alternative sources (a study from another group) should be proffered or alternatively an acknowledgement that the presence of appreciable plasma membrane ATPase contamination and, potentially other plasma membrane contamination (see following paragraphs), is a shortcoming of the approach taken. Moreover, additional organelle protein contamination in the authors’ preparations is still not discussed by the authors within the manuscript.

> The Western Blot analysis presented in the novel supplementary Figure 1a demonstrates the purity of our mitochondrial isolations and our analysis of the typical organellar markers in Table R2 further validates the purity of our material. We indeed discuss in the manuscript (page 5) that some of the identified proteins might represent contaminants (or are multiple localized mitochondrial proteins). We now included additionally the possibility of contaminations in the Methods section. Furthermore, we provide the reader with the valuable Yeast/Mito ratios for each protein (to allow judgement of a potential multiple localized protein or contamination for each protein). However, it is simply not possible to clarify this question experimentally for each of the proteins in a single study. The presence of contaminants is unavoidable in proteomic studies. We also would like to point out that we did not omit any identified proteins from our list of identified proteins including Pma1 and Pma2 (Suppl. Data 2). Moreover, as also pointed out below and in the manuscript, an increasing number of mitochondrial proteins are localized in several cellular compartments. E.g. we have recently identified Yck2 (a so far plasma membrane annotated protein to which the reviewer also refers to below) and its partner Yck1 to be localized to the mitochondrial membrane in addition to the plasma membrane (Gerbeth et al., *Cell Metabolism* 18, 578-587, 2013). This was a highly surprising and unexpected discovery (of a dual protein localization in the plasma membrane and mitochondrial membrane) which would not have been possible without previous proteomic studies (the plasma membrane annotated Yck1 was identified in two different mitochondrial proteomic studies (Reinders, *JPR* (2007) and Renvoise, *J. Proteomics* (2014)))

The fact that this information (i.e. annotation of Yck2 as mitochondrial protein) is not listed in the common databases (e.g. SGD) further demonstrates the necessity and impact of our present study.

The method used in the paper for classification of proteins to specific clusters in New Supplementary Figures 4 and 5 risks overfitting of the data as the entirety of training data is used to give a static view of the mitochondrial map. There is no assessment of the performance of the method used. The authors may consider leaving some data out of the training data from New Supplementary Figure 4 and seeing how this affects the classification obtained in New Supplementary Figure 5 and whether the proteins are assigned to the same clusters, or whether their model can assign the left out training data to the expected sub-organellar cluster.

> The statistical approach undertaken here (on request of the reviewer) showed a remarkable agreement with the original empirical data evaluation.

The dataset was trained with the original and highly solid reference set which was used from the very beginning of our study. The correlation of the data are now implemented in the Landscape dataset and allows the reader to compare both approaches (which overlap in 95% of the data).

The datapoints within the training set are well separated within their classes. Well separated means the three classes have a convex hull (i.e. ellipse) each which are almost not overlapping. Hence, we decided to do the analysis without measuring the performance measures (e.g. accuracy). We do not run into the risk of overfitting the model as this would imply that we used irrelevant features. Since we have only two features and more than 20x more data points than features, there is no danger of overfitting.

It is clear, from Supplementary Figure S2 of the originally submitted manuscript and the observations made in the paper (such as in New Supplementary Figure 7), that this method is successful in assigning a lot of proteins which are annotated in the literature as mitochondrial. However, the two discounted contaminants (Pma1 and Pma2, the plasma membrane ATPases), are “assigned” by this method to being part of the integral membrane cluster. These are clearly not integral to any mitochondrial membrane and yet are still assigned as such by this method. A thorough analysis of false discoveries in the results of this study needs to be undertaken so that it is clear what proportion of these observations are actually applicable to mitochondria. The authors must comment on false discoveries such as those already acknowledged (Pma1 and Pma2), and other contaminants in specified clusters (annotation mentioned is manually curated Cellular Component annotation from the Saccharomyces Genome Database), such as those listed below:

In the “soluble cluster”

Nucleoporin NUP116 (nuclear annotation).

AMS1 (vacuolar annotation).

In the “integral cluster”

IST2 and TCB3, both annotated as cortical ER and involved in cortical

ER/plasma membrane tethering.

HXT2, a high affinity hexose transporter of the plasma membrane.

GAS5 and EXG2, which are both annotated as being involved in plasma membrane and cell wall-related pathways and are annotated as being localised to these locations.

YCK2, which is annotated as localising to the bud tip, mating projection and plasma membrane.

JEN1, which is annotated as localising to the plasma membrane.

> It is beyond the scope of a proteomic study such as ours to discuss every single protein identified. Again, as outlined above: Yck2 (so far annotated as plasma membrane protein) was identified to reside also in the mitochondrial membrane (as integral protein) where it functions in the regulation of the mitochondrial import receptor Tom22 (Gerbeth et al., Cell Metabolism 18, 578-587, 2013). This is yet another example of an inaccurate or incomplete entry in the Saccharomyces Genome database. The fact that some of these entries are manually curated as pointed out by the reviewer does not exclude a further localization within the cell, in the case of Yck2 (and also its partner Yck1) in mitochondria. Moreover, two other proteins that the reviewer lists, Tcb3 and Jen1, are indeed annotated as mitochondrial in SGD (based on two different mitochondrial proteomic studies), supporting a possible multiple localization of these proteins including mitochondria.

Furthermore, it is not possible to assess a statistically reliable false discovery rate for our data set and supplying it would be providing a wrong impression to the reader ship. False discovery rates require a certain number of data points in order to allow a reasonable resolution. Classically, in proteomics the FDR is used to assess the number of random protein/peptide/spectrum identifications. This has been applied extensively in this study and relies on the so-called target/decoy model that allows estimating the amount of random spectrum identifications in the total dataset. The more target (true) identifications are present per decoy (false) hit, the more robust FDR estimates can be produced. However, for estimating a reliable FDR for the here proposed three clusters, which would predict the share of random (wrong) assignments to one of these clusters, would require a completely novel mathematical model for which there is no foundation. Whereas for protein identifications this is straightforward by using either reversed or randomized protein sequences as decoy, here there is no such straightforward approach. However, since we (i) only assigned proteins that were repeatedly quantified with at least 2 unique peptides in the different replicates, (ii) have demonstrated the high purity and intactness of mitochondria by multiple and partially complementary approaches, and (iii) can clearly show that our assignments fit perfectly to numerous protein complexes comprising different subcompartments and protein classes, including high and low abundant proteins, (iv) have verified the correct assignment for multiple well-known and novel proteins, and (iv) even can strengthen this using the now added mathematical model, which even provides likelihoods for the assignment of each protein to one of the different clusters, we are confident, that this dataset - even without an

FDR assessment, which in our opinion would not be reliable anyway – meets the high quality standards of both the proteomics and the mitochondria communities.

At the moment, this model seems to assign both mitochondrial and contaminant integral membrane proteins to the “integral” cluster, and contaminant proteins to the soluble cluster. It is conceivable that proteins such as those mentioned above may represent novel mitochondrial proteins, but they may also represent mis-assignments and false discoveries brought about as a direct consequence of the method used in the paper to prepare the mitochondria. The authors should therefore discuss the shortcomings of the method they used for classification of proteins to each of the clusters and how they dealt with any false discoveries. Although proteins from complexes such as those displayed in New Supplementary Figure 7 are found in the expected clusters, demonstrating that the method used works to an extent, protein assignments which conflict with the literature, such as those mentioned above should not be ignored. The authors should be encouraged to perform validation of some of their specific localisation predictions such as those potential novel localisations/false discoveries mentioned above. This may be performed using other means such as the import assay described by the authors or indirect immunofluorescence, due to the authors’ aforementioned interference of a fluorescent tag with mitochondrial protein import.

> The protein assemblies and machineries analyzed in Figure 2 (the enzymes of the citric acid cycle, Complex IV of the respiratory chain, the TIM import machineries, Coenzyme q biosynthesis) and new Supplementary Figure 6 (ATPase, MICOS complex, TOM, SAM and MIM machinery) and Rebuttal Figure R3 (additionally complex III of the respiratory chain) show a very strong correlation of our data with the published known mitochondrial sublocalizations and demonstrate adequately the quality of our dataset. These systems were chosen because of their well-known topologies, shown by various groups of scientists over the last decades in mitochondrial research. They represent all clusters assigned in our study and comprise high and low abundant mitochondrial proteins. Using still very vaguely studied proteins and their topologies within the mitochondrial subcompartments is in our opinion not suitable for the validation of our data.

Reviewer 3

I have no further concerns, and I congratulate the authors on an excellent manuscript that will be of great utility to the field.

> We thank the reviewer for this very positive comment.